# Diversification of multipotential postmitotic mouse retinal ganglion cell precursors into discrete types

Karthik Shekhar[1,2,3,4]*, Irene E Whitney[4†], Salwan Butrus[1], Yi-Rong Peng[4,5], Joshua R Sanes[4]*

[1]Department of Chemical and Biomolecular Engineering; Helen Wills Neuroscience Institute; Center for Computational Biology; California Institute for Quantitative Biosciences, QB3, University of California, Berkeley, Berkeley, United States; [2]Biological Systems and Engineering Division, Lawrence Berkeley National Laboratory, Berkeley, United States; [3]Broad Institute of Harvard and MIT, Cambridge, United States; [4]Center for Brain Science and Department of Molecular and Cellular Biology, Harvard University, Cambridge, United States; [5]Department of Ophthalmology, Stein Eye Institute, UCLA David Geffen School of Medicine, Los Angeles, United States

*For correspondence:
kshekhar@berkeley.edu (KS);
sanesj@mcb.harvard.edu (JRS)

Present address: †Honeycomb Biotechnologies, Waltham MA, United States

Competing interest: The authors declare that no competing interests exist.

**Abstract** The genesis of broad neuronal classes from multipotential neural progenitor cells has been extensively studied, but less is known about the diversification of a single neuronal class into multiple types. We used single-cell RNA-seq to study how newly born (postmitotic) mouse retinal ganglion cell (RGC) precursors diversify into ~45 discrete types. Computational analysis provides evidence that RGC transcriptomic type identity is not specified at mitotic exit, but acquired by gradual, asynchronous restriction of postmitotic multipotential precursors. Some types are not identifiable until a week after they are generated. Immature RGCs may be specified to project ipsilaterally or contralaterally to the rest of the brain before their type identity emerges. Optimal transport inference identifies groups of RGC precursors with largely nonoverlapping fates, distinguished by selectively expressed transcription factors that could act as fate determinants. Our study provides a framework for investigating the molecular diversification of discrete types within a neuronal class.

## Editor's evaluation

Your study using single-cell RNA-seq to profile developing retinal ganglion cells from embryonic and postnatal mouse retina showcases the diversification of this class of neuron into specific subtypes. The computational approaches you developed identify groups of RGC precursors with largely nonoverlapping fates, distinguished by selectively expressed transcription factors that could act as fate determinants. You then show that over time clusters of cells become 'decoupled' as they split into subclusters, indicating that subtype diversification arises as a gradual, asynchronous fate restriction of postmitotic multipotential precursors. Your data should enable the neural development community to generate new hypotheses in the field of retinal ganglion cell differentiation and beyond in other neural structures.

## Introduction

A central question in developmental neurobiology is how the brain's diverse neuronal types arise from multipotential progenitors (*Lodato and Arlotta, 2015*; *McConnell, 1991*; *Wamsley and Fishell,*

*2017*). The vertebrate retina has been a valuable model for addressing this question: it is about as complicated as any other region of the brain, but has several features that facilitate mechanistic analysis (*Dowling, 2012*). The retina contains five classes of neurons – photoreceptors that sense light, three interneuronal classes (horizontal, bipolar, and amacrine cells) that process visual information, and retinal ganglion cells (RGCs) that pass the information to the rest of the brain through the optic nerve (*Figure 1a*; *Masland, 2012*). These classes can be divided into numerous types, ~130 in mouse and chick, each of which has characteristic morphological, physiological, and molecular properties, and plays distinct roles in information processing (*Baden et al., 2016*; *Franke et al., 2017*; *Goetz et al., 2021*; *Macosko et al., 2015*; *Rheaume et al., 2018*; *Shekhar et al., 2016*; *Shekhar and Sanes, 2021*; *Tran et al., 2019*; *Yamagata et al., 2021*; *Yan et al., 2020*). Remarkably, nearly all types are distributed across the entire retina (*Kay et al., 2012*; *Keeley et al., 2020*; *Rockhill et al., 2000*), so morphogen gradients, which play a critical role in other parts of the central nervous system (e.g., *Sagner and Briscoe, 2019*), cannot provide an explanation for retinal neuronal diversification (*Marquardt and Gruss, 2002*).

Seminal studies have provided deep insights into how retinal classes arise (*Bassett and Wallace, 2012*; *Cepko, 2014*). First, lineage tracing in rodents and frogs showed that single retinal progenitor cells (RPCs) can give rise to neurons of all classes as well as glia, and are therefore multipotential (*Holt et al., 1988*; *Turner and Cepko, 1987*; *Turner et al., 1990*; *Wetts and Fraser, 1988*). Second, the competence of multipotential RPCs to generate cells of particular classes changes over time, accounting for their sequential (but overlapping) birth windows (*Cepko, 2014*; *Livesey and Cepko, 2001*). Such segregation of birth windows is a hallmark of many neuronal systems (*Holguera and Desplan, 2018*) and is believed to arise from the differential temporal regulation of gene expression in RPCs (*Blackshaw et al., 2004*; *Brown et al., 2001*; *Chen et al., 1997*; *Clark et al., 2019*; *Trimarchi et al., 2008*). Third, competence is probabilistic rather than deterministic, with stochastic factors accounting for variations in the distribution of cell classes generated by individual RPCs (*Boije et al., 2014*; *Gomes et al., 2011*; *Johnston and Desplan, 2010*).

In contrast to these well-established tenets of neuronal class generation, we know far less about how immature postmitotic neurons (which we call neuronal precursors here) committed to a specific class identity diversify into distinct types. We address this issue here, focusing on RGCs. All RGCs are similar in many respects: for example, they all elaborate dendrites that receive input from amacrine and bipolar interneurons, send axons through the optic nerve, and use glutamate as a neurotransmitter (*Sanes and Masland, 2015*). However, they differ in molecular, morphological, and physiological details, which have led to their division into ~45 distinct types in mice (*Baden et al., 2016*; *Bae et al., 2018*; *Goetz et al., 2021*; *Rheaume et al., 2018*; *Tran et al., 2019*). Most of these types appear to be feature detectors that collectively transmit a diverse set of highly processed images of the visual world to the rest of the brain (*Baden et al., 2020*; *Sanes and Masland, 2015*). Several genes have been implicated in maturation of a few mouse RGC types (*Clark et al., 2019*; *Kiyama et al., 2019*; *Liu et al., 2018*; *Lo Giudice et al., 2019*; *Lyu and Mu, 2021*; *Peng et al., 2017*; *Sajgo et al., 2017*), but a comprehensive investigation of RGC diversification has been lacking.

To gain insight into how and when adult RGC types emerge, we used high-throughput single-cell RNA-seq (scRNA-seq) to profile RGC precursors during embryonic and postnatal life in mice. We find that the number and distinctiveness of molecularly defined groups of precursors increases with developmental age, implying that types arise by a gradual process rather than from ~45 committed precursor types. Using statistical inference approaches, we identify fate associations among immature RGCs as transcriptomically distinct types emerge. These analyses suggest a model in which types arise from multipotential precursors by a process of restriction that we term fate decoupling. The decoupling is gradual and asynchronous, resulting in different types emerging at different times. We also use markers of RGCs that project to contralateral or ipsilateral retinorecipient areas to subdivide each type by its projection pattern, leading to the conclusion that laterality may be specified prior to type identity is fixed. Together, our results provide both a model of RGC diversification and a computational framework that can be applied generally to analyze the diversification of closely related neuronal types within a class.

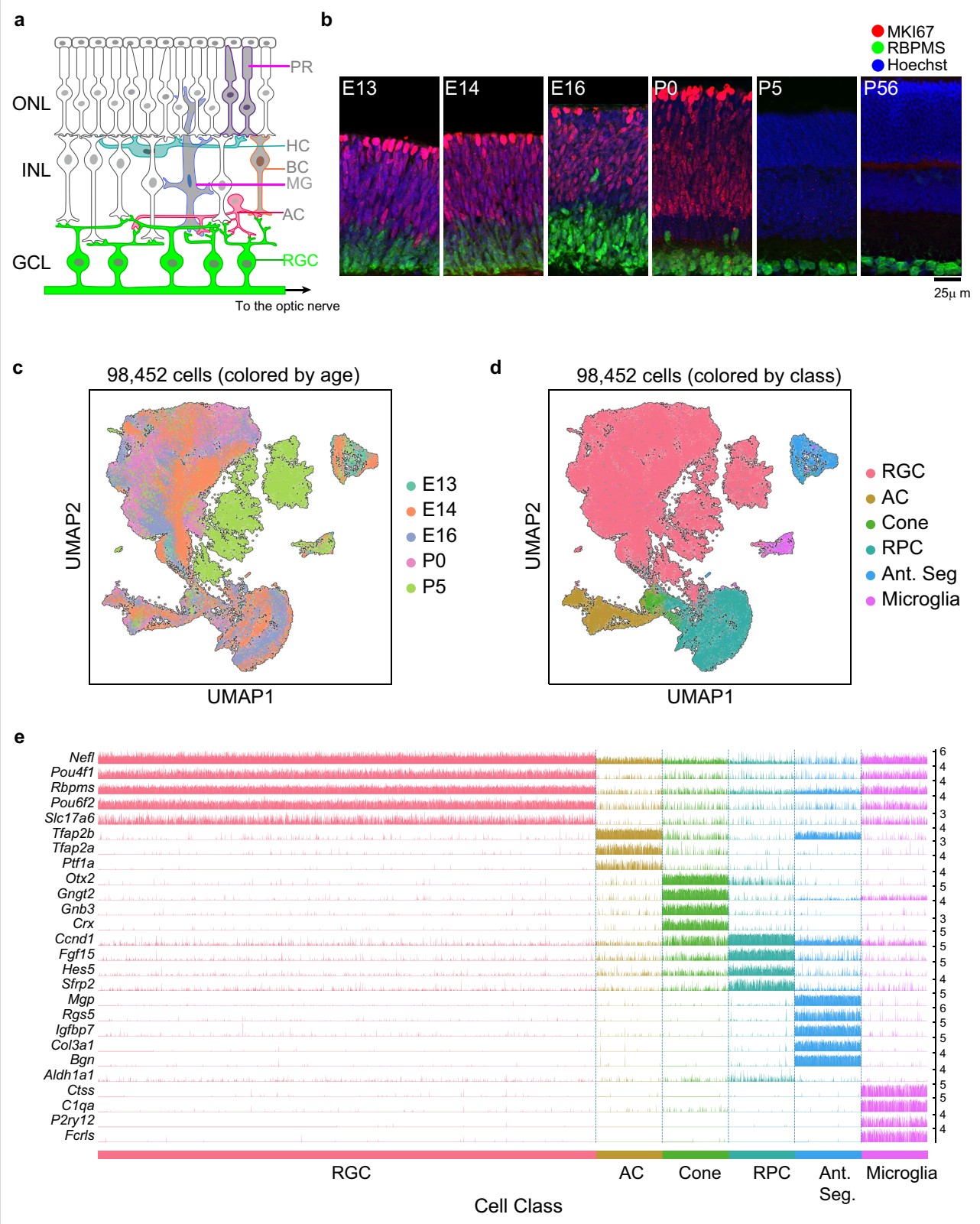

**Figure 1.** Transcriptomic profiling of single postmitotic retinal ganglion cells (RGCs) during embryonic and postnatal development in mice. (**a**) Sketch of a section of the mouse retina showing major cell classes – photoreceptors (PRs; rods and cones), horizontal cells (HCs), bipolar cells (BCs), amacrine cells (ACs), Müller glia (MGs), and RGCs. PRs reside in the outer nuclear layer (ONL), while BCs, HCs, and most ACs reside in the inner nuclear layer (INL). RGCs and some ACs reside in the ganglion cell layer (GCL). Axons of RGCs project to higher visual areas via the optic nerve. (**b**) Retinal section

*Figure 1 continued on next page*

Figure 1 continued

of the indicated ages labeled for the cell cycle marker *MKI67* (red) and the RGC marker *RBPMS* (green); nuclei are counterstained by the Hoeschst dye (blue). Micrographs are orientated as the schematic in panel (**a**). (**c**) Visualization of transcriptional diversity of 98,452 cells using Uniform Manifold Approximation and Projection (UMAP), a nonlinear dimensionality reduction algorithm that assigns proximal x-y coordinates to cells (dots) with similar transcriptional profiles (***Becht et al., 2018***). (**d**) Same as (**c**), with cells colored by cell class, assigned based on transcriptional signatures displayed in panel (**e**). RPC, retinal progenitor cells; Ant. Seg., anterior segment cells. (**e**) Trackplot showing expression patterns of cell class-specific marker genes (rows) across single cells (columns). Cells are grouped by class as in (**d**). For each class, we randomly sampled 20% of total cells covering all immature time points (embryonic day [E]13, E14, E16, postnatal day [P]0, P5). For each gene, the scale on the y-axis (right) corresponds to normalized, log-transformed transcript counts detected in each cell.

The online version of this article includes the following figure supplement(s) for figure 1:

**Figure supplement 1.** Separation of major transcriptomic groups and assessment of immature retinal ganglion cells (RGCs) at embryonic day (E)13 to postnatal day (P)5.

## Results

### Transcriptomic atlas of developing mouse RGCs

Mouse RGCs are born between approximately embryonic days (E) 11 and 17 with newborn RGCs exiting the mitotic cycle near the apical margin, then migrating basally to form the ganglion cell layer (***Dräger, 1985***; ***Marcucci et al., 2019***; ***Voinescu et al., 2009***; ***Figure 1b***). Reported birthdates differ among publications and are complicated by naturally occurring cell death and the central-peripheral developmental gradient, but a detailed analysis concludes that >95% of RGCs in adult mouse retina are born after E12.8 and >85% before E16 (***Farah and Easter, 2005***). Shortly after they are born, RGCs extend axons through the optic nerve, with some reaching retinorecipient areas by E15 (***Godement et al., 1984***; ***Osterhout et al., 2011***) and forming diverse projection patterns (***Martersteck et al., 2017***). During early postnatal life, they extend dendrites apically into the inner plexiform layer of the retina, receiving synapses from amacrine cells by postnatal day (P)4 and bipolar cells a few days later (***Kim et al., 2010***; ***Lefebvre et al., 2015***). Light responses are detected in RGCs by P10 but image-forming vision does not begin until eye-opening, around P14 (***Hooks and Chen, 2020***).

To determine when and how RGCs diversify, we used droplet-based scRNA-seq (***Macosko et al., 2015***; ***Zheng et al., 2017***) to profile them at five stages: E13 and E14 (during the period of peak RGC genesis), E16 (by which time RGCs axons are reaching target retinorecipient areas), P0 (as dendrite elaboration begins), and P5 (shortly after RGCs begin to receive synapses). As RGCs comprise ≤1% of retinal cells (***Jeon et al., 1998***), we enriched them with antibodies to two RGC-selective cell surface markers, Thy1/CD90 (***Barres et al., 1988***) and L1cam (***Demyanenko and Maness, 2003***; ***Figure 1—figure supplement 1a***).

We obtained a total of 98,452 single-cell transcriptomes with acceptable quality metrics (Materials and methods). Of these, we identified 75,115 (76%) as RGCs based on their expression of canonical RGC markers, including *Rbpms* (an RNA-binding protein) and *Slc17a6* (the vesicular glutamate transporter VGLUT2) (***Figure 1c–e***, ***Figure 1—figure supplement 1b and c***). Non-RGCs included amacrine cells (*Tfap2a+ Tfap2b+*), cone photoreceptors (*Otx2+ Crx+* ), microglia (*P2ry12+ C1qa+*), anterior segment cells (*Mgp+ Bgn+* ), and RPCs. Anterior segment cells were found only in E13 and E14 samples because whole eyes were dissociated at these stages. RPCs formed a continuum, containing both 'primary' RPCs expressing cell cycle-related genes (e.g., *Mki67, Ccnd5,* and *Birc5*) and previously described RPC regulators (e.g., *Sfrp2, Vsx2,* and *Fgf15*), and 'neurogenic' RPCs expressing proneural transcription factors (TFs) (e.g., *Hes6, Ascl1,* and *Neurog2*) (***Clark et al., 2019***). Importantly, these markers were not expressed in cells annotated as RGCs (***Figure 1—figure supplement 1d***). These stringent criteria ensured that our dataset comprised postmitotic committed RGCs, allowing us to focus on their diversification and maturation.

Overall, we recovered ~5900–18,500 RGCs at each of the five time points. Of the two surface markers used for enriching RGCs, Thy1 was effective at later stages as shown previously (***Kay et al., 2011***; ***Rheaume et al., 2018***; ***Tran et al., 2019***), whereas L1cam expression was more selective at E13 and E14 (***Figure 1—figure supplement 1b and c***). However, identical clusters were observed with both methods at E13, E14, and E16, albeit with different frequencies. This concordance supports the idea that neither marker fails to capture particular RGC types. To further evaluate the effectiveness of our enrichment strategy at early stages, we compared our data with two recent studies in which

developing retinal cells were profiled using scRNA-seq without any enrichment (*Clark et al., 2019*; *Lo Giudice et al., 2019*). A joint analysis of these datasets at embryonic time points showed consistency in the transcriptional signatures of major cell groups without discernible biases (*Figure 1—figure supplement 1e–g*). However, our enrichment protocols increased the fractional yield of RGCs by >3× at E14 and E16 and by >100× at P0 (*Figure 1—figure supplement 1h*), which enabled us to resolve heterogeneity within this class at immature stages. We also compared our P5 data with those from an earlier study in which P5 RGCs were profiled (*Rheaume et al., 2018*) and found a good correspondence (*Figure 1—figure supplement 1i*). For the analysis that follows, we combined precursor RGCs (E13–P5) with a previously described dataset of 35,699 mature RGCs at P56 (*Tran et al., 2019*).

## Immature RGCs diversify postmitotically

One can envision two extreme models of RGC diversification. In one, RGC type would be specified at or before mitotic exit, with each type arising from a distinct set of committed precursors that are transcriptomically defined. At the other extreme, all precursor RGCs would be identical when they exit mitosis and gradually acquire distinct identities as they mature (*Figure 2a*). Intermediate models could involve multiple groups of precursor RGCs, each biased towards a distinct set of terminal types.

To distinguish among these alternatives, we analyzed the transcriptomic diversity of RGCs at each developmental stage using the same dimensionality reduction and graph clustering approaches devised for analysis of adult RGCs (*Tran et al., 2019*; see Materials and methods). This analysis led to three main results.

First, RGCs were already heterogeneous soon after mitotic exit. There were 10 transcriptionally defined precursor clusters at E13 (*Figure 2b*) before or at the peak time of RGC birth. The number of discrete clusters increased only slightly by E14 (from 10 to 12; *Figure 2c*), arguing against a model in which the number of precursor types extrapolated back to 1. No single cluster dominated the frequency distribution at either time as would be expected if a totipotent precursor RGC were to exist shortly after terminal mitosis.

Second, the number of transcriptionally defined clusters increased gradually, between E13 and adulthood, reaching 45 only after P5 (*Figure 2b–g*). Several arguments indicate that this increase is biologically significant rather than being an artifact of the data or computational analysis. (1) We used the same clustering procedure at all ages. (2) The qualitative trends were robust against variations in clustering parameters. (3) All embryonic clusters contained cells isolated with both cell markers, L1cam and *Thy1* (*Figure 2—figure supplement 1a–c*), indicating that lower cluster numbers at early stages did not result from biased collection methods. (4) The increase in the number of effective molecular types was robust as demonstrated by three diversity indices – Rao, Simpson, and Shannon – all of which buffer against artificial inflation of diversity due to small clusters (*Figure 2h*, *Figure 2—figure supplement 1d*; see Materials and methods). (5) There was no systematic dependence of the number of clusters on the number of cells. For example, we identified 12 clusters from 17,100 cells at E14 and 38 clusters from 17,386 cells at P5.

Third, the transcriptomic variation became increasingly discrete with age. We quantified this increase in inter-cluster separation by calculating (1) the average cross-validation error of a multi-class classifier, and (2) the ratio of mean cluster diameter to mean inter-cluster distance in the low dimensional embedding (Materials and methods). Both metrics decrease in numerical value as the clusters are more well defined. From these trends, we conclude that the boundaries between RGC clusters become sharper as development proceeds (*Figure 2i and j*).

Taken together, our results show that transcriptomic clusters of RGCs increase in number and distinctiveness over time, making it unlikely that RGC-type identity is fully specified at the progenitor stage.

## Temporal relationships among immature RGC clusters

We next investigated the temporal relationships among precursor RGC clusters identified at different ages. We again consider two extreme models. In a 'specified' model, each terminal type arises from a single cluster at every preceding developmental stage (*Figure 3a*, left). In this model, distinct transcriptomic states among precursor RGCs correspond to distinct groups of fates. At the other extreme, distinct clusters would share similar sets of fates (*Figure 3a*, right). In an intermediate model, fates of precursor clusters would exhibit partial overlap.

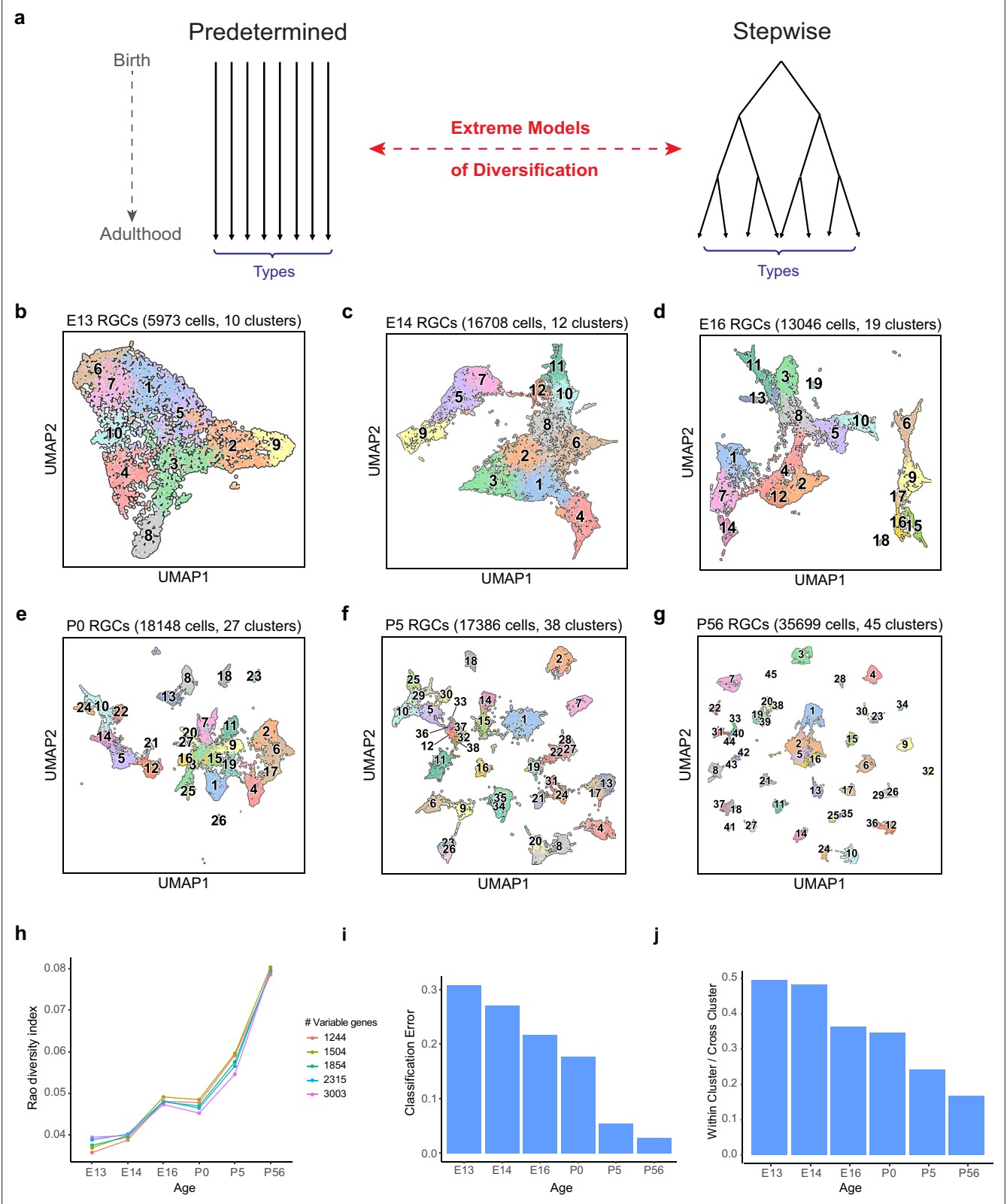

**Figure 2.** The number and discreteness of transcriptomic clusters of retinal ganglion cells (RGCs) increase with age. (**a**) Extreme models of RGC diversification. In one scenario (left), immature RGCs commit to one of the terminal types by the time of birth (i.e., mitotic exit) or shortly after. Alternatively (right), initially identical postmitotic RGC precursors acquire distinct molecular identities in a gradual process of restriction. (**b–g**) Visualization of transcriptomic diversity of immature RGCs at embryonic day (E)13 (**b**), E14 (**c**), E16 (**d**), postnatal day (P)0 (**e**), P5 (**f**), and P56 (**g**) using

*Figure 2 continued on next page*

*Figure 2 continued*

Uniform Manifold Approximation and Projection (UMAP). Cells are colored by their cluster identity, determined independently using dimensionality reduction and graph clustering (Materials and methods). Clusters are numbered based on decreasing size at each age. Data for adults (P56) are replotted from *Tran et al., 2019*. In that study, 45 transcriptomic types were identified via unsupervised approaches, one of which was mapped to two known functional types by supervised approaches. We do not distinguish them in this study. (**h**) Transcriptional diversity of RGCs as measured by the Rao diversity index (y-axis) increases with age (x-axis). The trend is insensitive to the number of genes used to compute inter-cluster distance (colors). See Materials and methods for details underlying the calculation. (**i**) Transcriptomic distinctions between RGC clusters become sharper with age as shown by decreasing average per-cluster error of a multiclass classifier with age. Gradient boosted decision trees (*Chen and Guestrin, 2016*) were trained on a subset of the data and applied on held-out samples to determine the test error. (**j**) RGC clusters also become better separated in the UMAP embedding, as shown by the decreasing values of the average relative cluster diameter with age.

The online version of this article includes the following figure supplement(s) for figure 2:

**Figure supplement 1.** Transcriptomic diversity of immature retinal ganglion cells (RGCs) by age.

As a first step in discriminating among these scenarios, we used transcriptome-wide correspondence among clusters as a proxy for fate association. We identified mappings between clusters across each pair of consecutive developmental stages (E13–E14, E14–E16, E16–P0, P0–P5, and P5–P56) using gradient boosted trees (*Chen and Guestrin, 2016*), a supervised classification approach (Materials and methods). In each case, a classifier trained on transcriptional clusters at the older stage was used to assign older cluster labels to cells at the younger stage (e.g., E16 labels assigned to E14 RGCs). Patterns expected for the extreme models are schematized as 'confusion matrices'(*Stehman, 1997*) in the lower panels of *Figure 3a*.

Correspondence fell between the two extremes (*Figure 3d–h*, *Figure 3—figure supplement 1a–d*). We quantified the extent of correspondence using two metrics: normalized conditional entropy (NCE) and the adjusted Rand Index (ARI) (Materials and methods). Both NCE and ARI are restricted to the range (0,1), with lower values of NCE and higher values of ARI consistent with a specified mode of diversification. Both metrics exhibited an increased degree of specificity with age (*Figure 3b and c*). Since NCE and ARI provide a single measure of specificity for the entire datasets being compared, we also computed a 'local metric,' the occupancy fraction (OF), which quantifies mapping specificity for each cluster (Materials and methods). Results based on this metric were consistent with increased specificity of correspondence with age (*Figure 3—figure supplement 1e*). Overall, this analysis of transcriptomic correspondence suggests that poorly specified relationships among transcriptomic clusters at early stages are gradually refined to yield increasingly specific associations at later stages.

## Immature RGCs are multipotential

The classification analysis presented so far relied on comparing clusters between ages and was therefore unable to link individual precursors to specific terminal fates. At one extreme, individual precursor clusters might contain several groups of cells, each committed to a distinct, small number of fates. Alternatively, individual cells might be as multipotential as the clusters in which they reside (*Figure 4a*).

Unfortunately, this approach does not afford a straightforward way to explore variations in patterns of fate associations within clusters. We therefore turned to Waddington optimal transport (WOT), a computational method rooted in optimal transport theory (*Kantorovich, 2006*; *Monge, 1781*) that utilizes scRNA-seq measurements at multiple stages to infer developmental relationships (*Schiebinger et al., 2019*). Briefly, WOT computes a 'transport matrix' $\Pi$ between each pair of consecutive ages with elements $\Pi_{ij}$ encoding fate associations between a single RGC $i$ at the younger age and RGC $j$ at the older age (see Materials and methods). WOT directly computes fate associations at the level of individual cells without requiring clustering as a prior step. We conducted computational tests to assess the numerical stability of associations reported by WOT (*Figure 4—figure supplement 1*). We also determined that when collapsed to the level of clusters the WOT-inferred transport maps strikingly mirrored the confusion matrices obtained from multi-class classification (*Figure 4—figure supplement 2*).

Based on the success of these tests, we applied WOT to compute the 'terminal fate' for each precursor RGC. We leveraged the fact that in WOT fate associations between RGCs at nonconsecutive ages (e.g., E16 and P56) can be estimated in a principled way by multiplying the intermediate transport matrices. This yielded a fate vector $\vec{f}$ for each of the 75,115 immature RGCs, whose $k$th element $f_k$ represents the predicted probability of commitment to adult type $k \in (1, 2, \ldots, 45)$ (Materials and

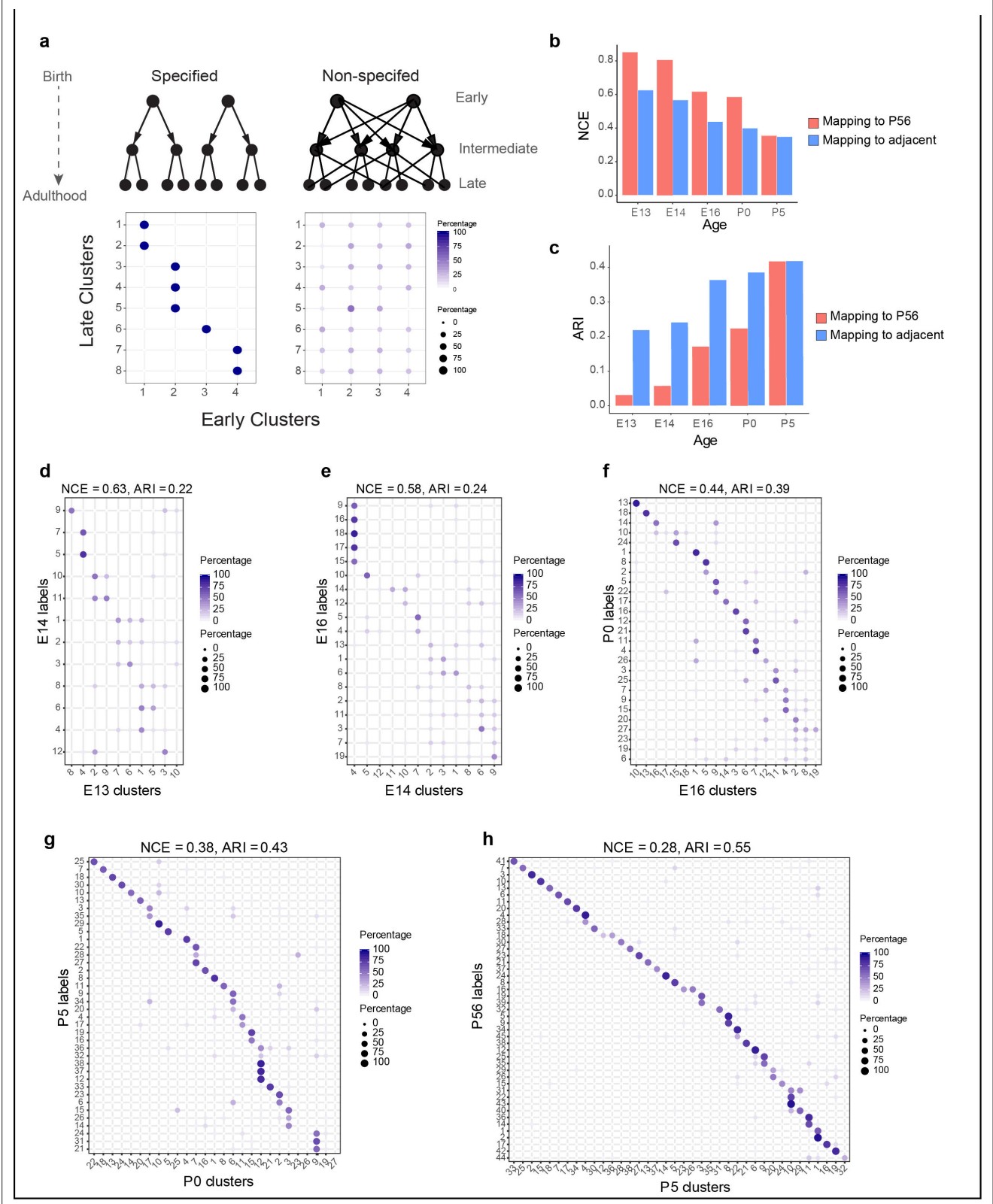

**Figure 3.** Incompletely specified temporal relationships among retinal ganglion cell (RGC) clusters. (**a**) Top: specified (left) and nonspecified (right) modes of diversification. Nodes denote transcriptomic clusters of immature RGCs, and arrows denote fate relationships. Bottom: confusion matrices depicting transcriptomic correspondence between late and early clusters expected for the two modes. Circles and colors indicate the percentage of a given late cluster (row) assigned to a corresponding early cluster (column) by transcriptome-based classifier trained on early clusters. The number of

*Figure 3 continued on next page*

*Figure 3 continued*

late and early clusters has been set to 8 and 4 for illustration purposes. (**b**) Barplot showing values of the normalized conditional entropy (NCE) for each age calculated using the transcriptional cluster IDs and the XGBoost-assigned cluster IDs corresponding to the next age or to postnatal day (P)56 (e.g., for embryonic day [E1]3, the NCE was calculated across E13 RGCs by comparing their transcriptional cluster ID with the assigned E14 cluster IDs based on a classifier trained on the E14 data). Lower values indicate specific mappings. (**c**) Same as (**b**), but plotting values of the adjusted Rand Index (ARI), where larger values correspond to higher specificity. (**d–h**) Confusion matrices (representation as in **a**), showing transcriptomic correspondence between consecutive ages: E14–E13 (**d**), E16–E14 (**e**), P0–E16 (**f**), P5–P0 (**g**), and P56–P5 (**h**). In each case, the classifier was trained on the late time point and applied to the early time point. Rows sum to 100%.

The online version of this article includes the following figure supplement(s) for figure 3:

**Figure supplement 1.** Temporal correspondences between transcriptomic clusters evaluated using supervised classification.

methods). A fully committed precursor would have all but one element of $\vec{f}$ equal to zero, whereas a partially committed precursor would have multiple nonzero elements in $\vec{f}$. Since the elements of $\vec{f}$ are interpreted as probabilities, they are normalized such that $\sum_k f_k = 1$.

We quantified the commitment of each precursor by computing its 'potential' $P = \frac{1}{\sum_k f_k^2}$, which is defined analogously to the 'inverse participation ratio' in physics (*Fyodorov and Mirlin, 1992*). In our case, the value of $P$ for a given RGC ranges continuously between 1 and 45, with lower values implying a commitment to specific fates and higher values reflecting indeterminacy. Importantly, this measure of commitment does not rely on arbitrary thresholding of the $f_k$ values to assign precursors to types.

Five results emerged from this analysis.

- Nearly all prenatal RGCs (i.e., on or before P0) were multipotential rather than committed to a single terminal fate, with individual potentials distributed across a range of values (*Figure 4b*).
- Multipotentiality was a general feature of immature RGCs, being present in cells of all clusters at E13, E14, and E16 (*Figure 4c–f*).
- At early stages, the average value of $P$ varied among transcriptomic clusters, reflecting asynchronous specification (*Figure 4c*). The tempo of commitment is further explored in the next section.
- Although they were multipotential, no precursor RGC was totipotential (i.e., completely unspecified, corresponding to $P = 45$). At E13, the average value of P was 11.6 ± 4.9, which was four-fold lower than the maximum possible value of 45, and no precursor had $P > 30$.
- Finally, inferred multipotentiality decreased gradually during development, and some persisted postnatally (average $P = 3.4 ± 2.1$ at P0, and 1.6 ± 0.9 at P5; *Figure 4g and h*).

From these results, we conclude that early postmitotic RGCs are multipotential but not totipotential, and that type identity is specified gradually via progressive restriction.

## Asynchronous specification of mouse RGC types via fate decoupling

As a first step in understanding the progressive restriction of RGC fate, we analyzed the extent to which pairs of mature types were likely to have arisen from the same set of immature precursors. To this end, we computed a 'fate coupling' value $C(l, m; age)$ for each pair of terminal RGC types ($l$ and $m$), defined as the Pearson correlation coefficient between the values of $f_l$ and $f_m$ across all precursors at a given age (Materials and methods). $f_l$ and $f_m$ are fate probabilities corresponding to types $l$ and $m$ as defined in the previous section. Values of $C(l, m; age)$ in our data ranged from −0.11 to 0.95. Higher values of $C(l, m; age)$ indicate strong fate coupling between types $l$ and $m$, implying the existence of common postmitotic precursors, whereas low $C(l, m; age)$ values suggest that types $l$ and $m$ arose from largely nonoverlapping sets of precursors. We visualized the pattern of fate couplings as network graphs, where the nodes represent types and the edge weights represent values of $C(l, m; age)$. The arrangement of nodes was determined at E13 using a force-directed layout algorithm (*Fruchterman and Reingold, 1991*), with pairwise distances being inversely proportional to the values of $C(l, m; E13)$, the fate coupling values at E13 (*Figure 5a*). To visualize the temporal evolution of these fate couplings, we retained the same layout of nodes while updating edge weights according to $C(l, m; age)$ (*Figure 5b–e*).

Types that were coupled in fate at the earliest time point gradually decoupled as development proceed. For example, at E13, 118/990 pairs (12%) were strongly coupled (threshold of $C(l, m; age) > 0.2$ as determined by randomization tests; see Materials and methods), while at P5, only 8/990 (<1%)

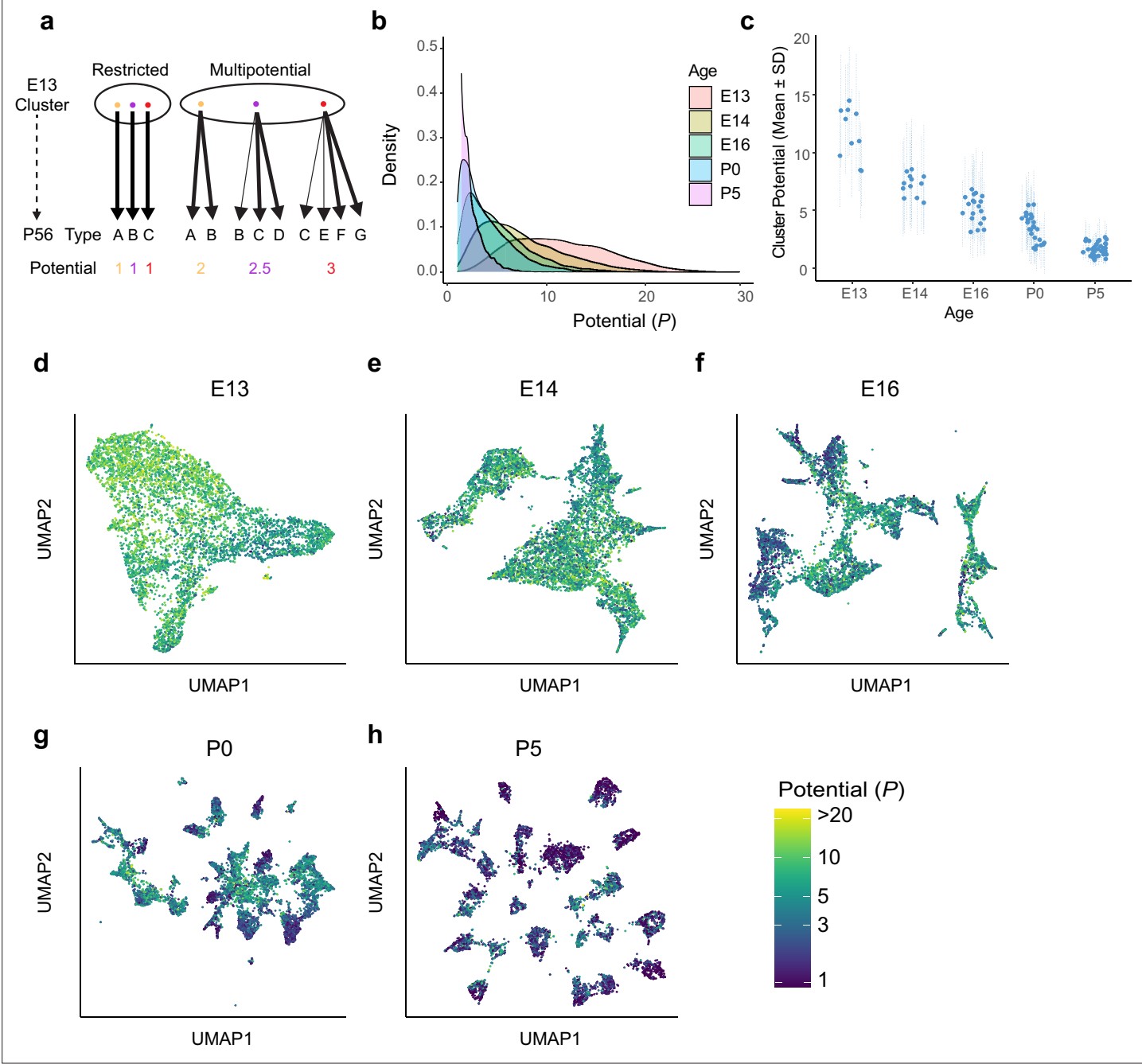

**Figure 4.** Multipotential fate associations between immature retinal ganglion cells (RGCs) and terminal types inferred via optimal transport. (**a**) Extreme models of diversification at single-cell resolution. Multipotential fate associations in a transcriptionally defined cluster (ellipse) could arise from a mixture of unipotential RGCs (left) or from multipotential RGCs (right). (**b**) Distributions of potential *P* across immature RGCs by age showing that restriction increases with age. (**c**) Inter- and intra-cluster variation of potential by age. At each age, variation in the potential values is shown for each transcriptomically defined cluster at that age. Dots denote the average potential, and dotted lines depict the standard deviation for cells within each cluster. (**d–h**) Uniform Manifold Approximation and Projection (UMAP) projections of embryonic day (E)13 (**d**), E14 (**e**), E16 (**f**), postnatal day (P)0 (**g**), and P5 (**h**) RGCs as in *Figure 2*, but with individual cells colored by their inferred potential. Potential of all RGCs at P56 = 1. The colorbar on the lower right is common to all panels, and values are thresholded at *P* = 20.

The online version of this article includes the following figure supplement(s) for figure 4:

**Figure supplement 1.** Variations in Waddington optimal transport (WOT)-inferred temporal couplings and tests across variations in hyperparameters.

**Figure supplement 2.** Temporal correspondences between transcriptomic clusters evaluated using Waddington optimal transport (WOT).

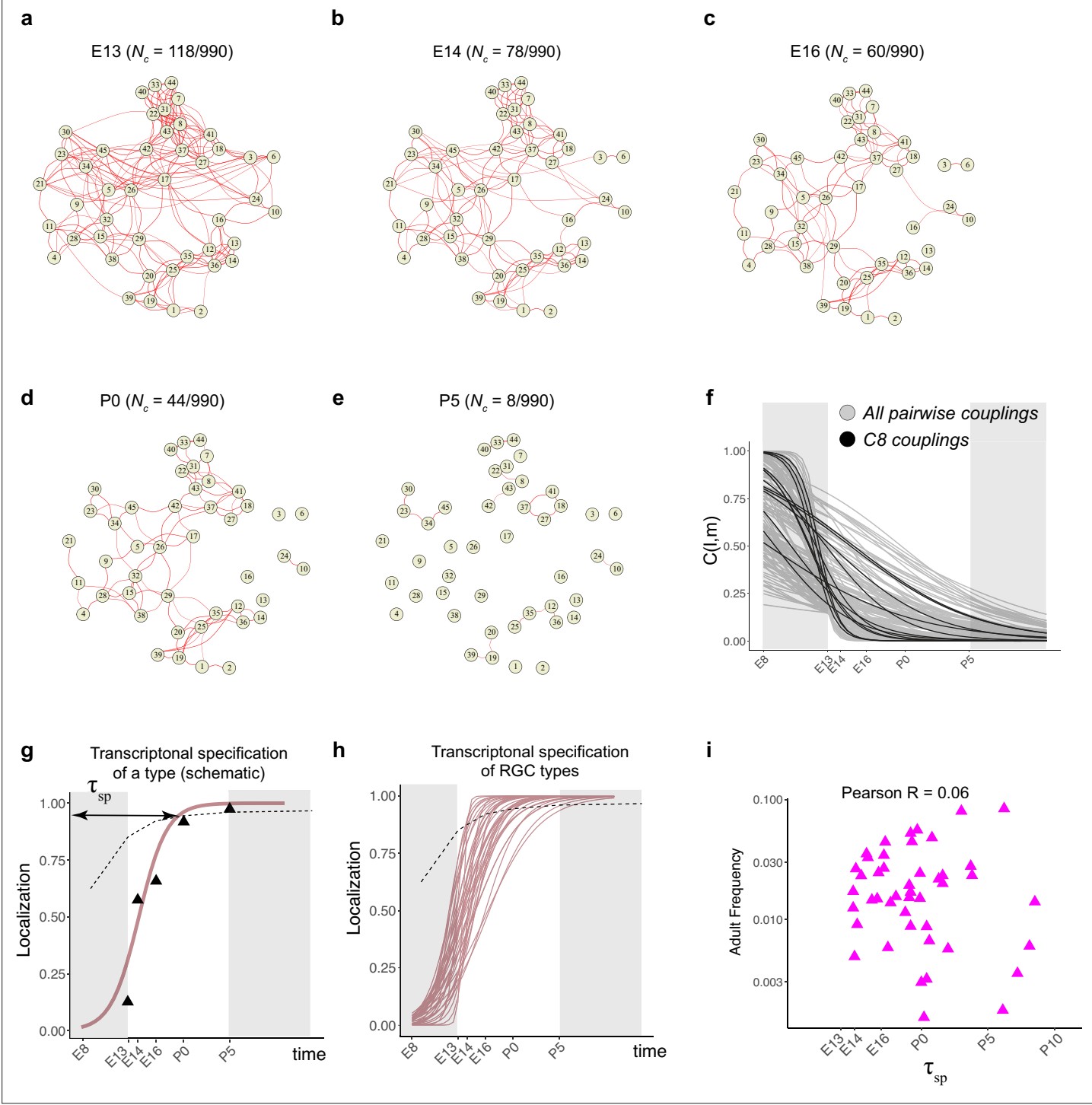

**Figure 5.** Fate decoupling of retinal ganglion cell (RGC) types. (**a**) Force-directed layout visualization of fate couplings at embryonic day (E)13, with nodes representing RGC types (numbered as in *Tran et al., 2019*) and the thickness of edges representing values of *C(l,m;E13)*. Edges with *C(l,m; E13)* < 0.2 are not shown. Number of edges with *C(l,m; E13)* > 0.2 are indicated on top. (**b–e**) Visualization of fate couplings at E14 (**b**), E16 (**c**), postnatal day (P)0 (**d**), and P5 (**e**). The positions of the nodes are maintained as in panel (**a**), but the edges are redrawn based on values of *C(l,m;age)* at each age. As in panel (**a**), we only show edges *C(l,m; age)* > 0.2. (**f**) The decay of pairwise fate couplings (y-axis) with age (x-axis). Each line corresponds to the temporal decay of *C(l,m)* for RGC pair l and m estimated via a logistic model (Materials and methods). For each pair, couplings at each age were fit to a model $C\left(l,m;age\right) = 1/\left(1 + e^{\beta_0 + \beta_1 * age}\right)$ with $\beta_0$, $\beta_1$ representing fitted parameters. The fitting was performed using data for ages E13, E14, E16, P0, and P5. The shaded portions correspond to the periods E8–E13 and P5 representing the extrapolations of the model. Black lines highlight the decay of all

*Figure 5 continued on next page*

*Figure 5 continued*

nonzero pairwise couplings for RGC type *C8* as an example. (**g**) Schematic showing logistic modeling to estimate specification time $\tau_{sp}$ for a particular type. The y-axis is a measure of the extent to which precursors biased towards the type are present in a single transcriptomically defined cluster (i.e., localization, see Materials and methods for details). Localization is defined as a numerical value in the range (0, 1) with higher values consistent with increasing specification. Individual triangles represent the localization values computed using Waddington optimal transport (WOT)-inferred fate couplings at each age, while the curve represents the fit using the logistic model. Dotted line shows the minimum threshold a type to be specified at each age. Its curved shape arises due to the increase in the number of clusters with age. (**h**) Localization curves (as in **g**) for the 38 RGC types showing the range of inferred specification times. Seven low-frequency types have been excluded from display (see *Figure 5—figure supplement 1d*). (**i**) Scatter plot showing poor correlation between adult frequency of a type (from *Tran et al., 2019*) and its predicated specification time (calculated from **h**).

The online version of this article includes the following figure supplement(s) for figure 5:

**Figure supplement 1.** Fate decoupling and temporal specification of retinal ganglion cell (RGC) types.

passed this criterion (*Figure 5a and e*). Lowering this threshold for coupling to 0.05 increased the number of strongly coupled pairs at P5 to only 2% (20/990).

Different pairs of types decoupled at different rates (*Figure 5f*). As they decoupled, RGC precursors became increasingly restricted to a single type (i.e., $f_k \gg f_{l \neq k}$ for a precursor favoring type *k*). This corresponded to a 'localization' of precursors in transcriptomic space and is a proxy for specification (see Materials and methods). We modeled the extent of localization vs. age via a logistic function (*Figure 5g*, *Figure 5—figure supplement 1d*) and used this to calculate a specification time for each type ($\tau_{sp}$) (see Materials and methods for details). Based on this analysis, 7/45 types are specified postnatally. The average $\tau_{sp}$ for RGCs was E17.8, but individual RGC types exhibited a wide range from E13.9 to P5.2 (*Figure 5h*). The inferred specification time was not correlated with adult frequency (*Figure 5i*).

We illustrate this range by considering three pairs of RGC types in *Figure 5—figure supplement 1*. C12 and C22 (numbered as in *Tran et al., 2019*; see *Figure 2g*) exhibit low fate coupling at all ages profiled (*Figure 5—figure supplement 1a*), indicative of separate precursor populations. In contrast, C19 and C20 decouple only at P0, implying the existence of a common precursor throughout embryogenesis (*Figure 5—figure supplement 1b*). C21 and C34 display an intermediate pattern, decoupling around E16 (*Figure 5—figure supplement 1c*). Taken together, these results suggest that RGC types emerge by asynchronous fate decoupling of multipotential precursors.

## Fate decoupled groups of RGC types defined by transcription factors

Because fate coupling is a metric of inferred overlap of developmental history, it is likely that tightly coupled types share common precursors. This relationship implies that tightly coupled types might also be specified by common transcriptional programs. As a step towards identifying candidate fate determinants, we identified eight TFs that are expressed by distinct groups of mature RGC types (*Figure 6a*, *Figure 6—figure supplement 1a*). Three of these are well-characterized RGC-selective TFs: *Foxp2*, expressed by five F-RGC types (*Rousso et al., 2016*); *Tbr1*, expressed by five T-RGC types (*Liu et al., 2018*); and *Eomes* (also known as *Tbr2*), expressed by seven types (*Mao et al., 2020*; *Tran et al., 2019*). The seven Eomes/Tbr2 types include the melanopsin-expressing intrinsically photosensitive (ip) RGC types (*Berson et al., 2002*). The remaining five were *Neurod2*, *Irx3*, *Mafb*, *Tfap2d*, and *Bnc2*, which label 8, 5, 4, 6, and 3 types, respectively. *Eomes* types also co-expressed *Tbx20* and *Dmrbt1* while *Neurod2* types also co-express *Satb2*. Together, 40/45 mature types expressed at least one of these TFs in a manner that was, with a few exceptions, mutually exclusive. In many cases, the fate proximity of types that shared TF expression was obvious (*Figure 6a*).

We refer to these TF-based groups as fate-restricted RGC subclasses – an intermediate taxonomic level between class and type based on inferred fate relationships. Consistent with their definition, the pairwise fate coupling among types from different subclasses was significantly lower than among types from the same subclass (*Figure 6b*). Thus, precursor RGC states associated with any two subclasses are more distinct than those associated with any two types. This is evident by the significant separation at E13 and negligible overlap at P5 for precursors favoring the *Eomes*, *Mafb*, and *Neurod2* subclasses, respectively, as shown in *Figure 6c–e*.

We also asked whether the TF-based subclasses differed in inferred transcriptomic specification time $\tau_{sp}$, as defined in *Figure 5g*. As shown in *Figure 6f–h* and *Figure 6—figure supplement 1b–e*, four subclasses were specified within a narrow interval (E16.8–E17.2), but three others differed

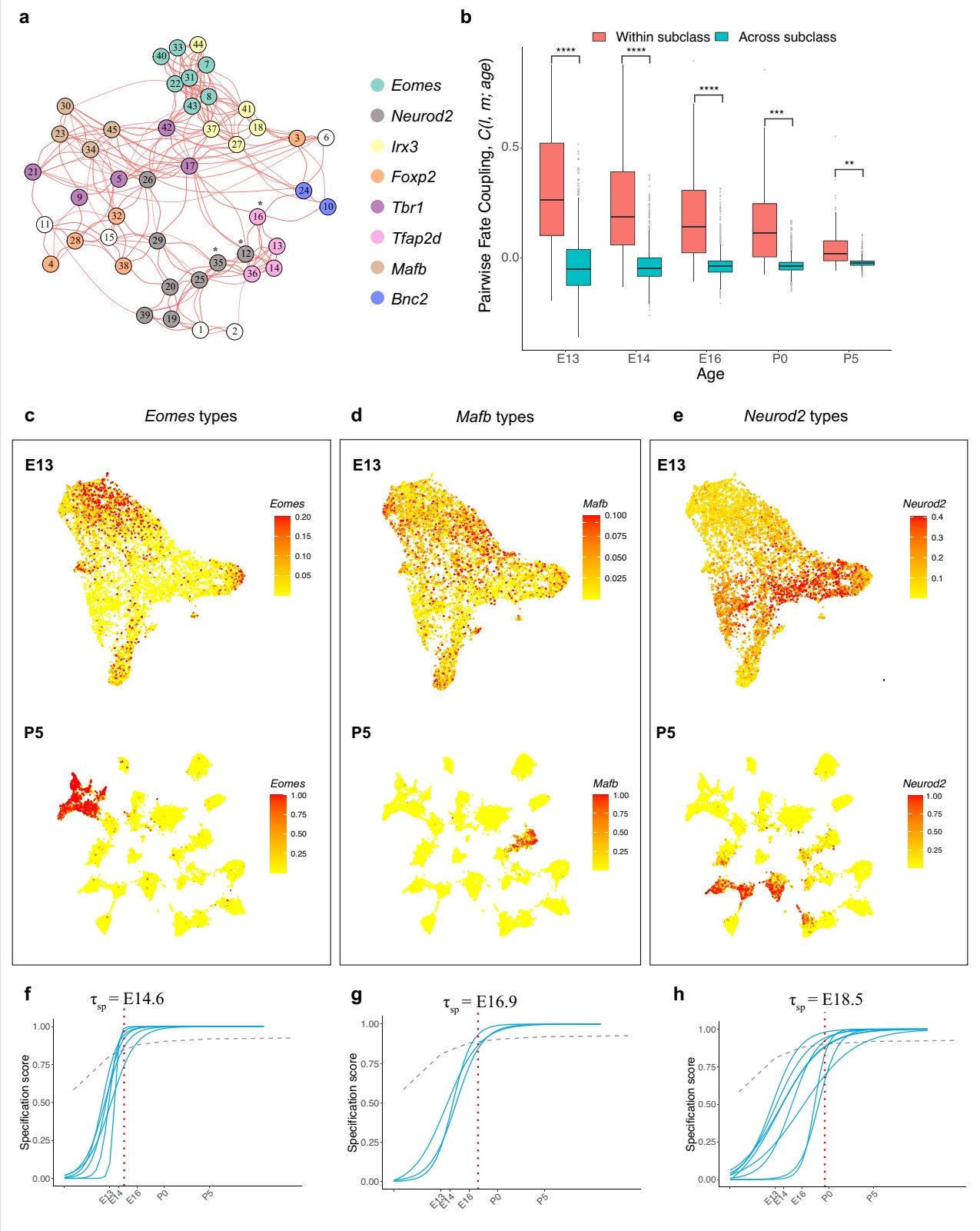

**Figure 6.** Temporal dynamics of retinal ganglion cell (RGC) subsets expressing specific transcription factors (TFs). (**a**) Embryonic day (E)13 network graph of fate couplings from *Figure 5a*, with RGC types colored based on their selective expression of TFs at postnatal day (P)56. Asterisks denote 3/45 types that express more than one TF (also see *Figure 6—figure supplement 1a*). (**b**) Box-and-whisker plots showing that pairwise fate couplings are higher between types within the same TF subclass than between types in different TF subclasses at all immature ages. Black horizontal line, median; bars,

*Figure 6 continued on next page*

*Figure 6 continued*

interquartile range; vertical lines, 1st and 99th percentile; dots, outliers. Asterisks indicate significant p-values based on a two-sided *t*-test (****p<10⁻⁷; ***p<10⁻⁵; **p<10⁻²). (**c**) *Eomes+* types. Top: Uniform Manifold Approximation and Projection (UMAP) representation of E13 RGCs with cells colored based on their cumulative fate association towards the seven *Eomes+* types. Bottom: UMAP representation of P5 RGCs with cells colored based on their cumulative fate association towards the seven *Eomes+* types. The value corresponding to the color of each cell (colorbar, right) can be interpreted as the probability of commitment towards the corresponding subclass. Note that the color does not denote the expression level for the gene. (**d**) Same as (**c**) for *Mafb+* types. (**e**) Same as (**c**) for *Neurod2+* types (**f–h**). Localization curves (as in **Figure 5g**) for *Eomes+* types (**f**), *Mafb+* types (**g**), and *Neurod2+* types (**h**). The mean inferred specification time $\tau_{sp}$ for each group is indicated on the top of each panel.

The online version of this article includes the following figure supplement(s) for figure 6:

**Figure supplement 1.** Transcription factor (TF)-based subgroups.

substantially. The average specification time for the *Eomes* group was E14.6 (p<0.0001, Student's *t*-test, compared to the mean for all types), while that for the *Mafb* and *Neurod2* groups were E16.9 (p<0.001) and E18.5 (p<0.0001), respectively. The early specification of the *Eomes* group is consistent with birthdating studies showing the average earlier birthdate of ipRGCs compared to all RGCs (**McNeill et al., 2011**).

In summary, our results suggest the existence of fate-restricted RGC subclasses that arise from distinct sets of precursors and diversify into individual types. This method of defining RGC groups, which relies on inferred proximity of precursors in transcriptomic space, is distinct from previous definitions of RGC subclass based on shared patterns of adult morphology, physiology, or gene expression (see Discussion). Accordingly, the fate couplings at E13 were only weakly correlated with transcriptomic proximity in the adult retina (**Figure 6—figure supplement 1f**). Further, while TF-based groups align with some previously defined subclasses (e.g., ipRGCs or *Tbr1+* RGCs), they do not map to other subclasses such as alpha-RGCs (four types) or T5-RGCs (nine types) (**Figure 6—figure supplement 1g**).

## Transcriptomic profiles of ipsilateral and contralateral RGCs

Finally, we considered the origin of two RGC groups defined by their projections: those with axons that remain ipsilateral at the optic chiasm (I-RGCs) and those that cross the midline to innervate contralateral brain structures (C-RGCs). The proportion of I-RGCs varies among vertebrates, in rough correspondence to the extent of binocular vision, ranging from none in most lower vertebrates to ~50% in primates. In mice, 3–5% of RGC axons remain ipsilateral, with most I-RGCs residing in the ventrotemporal (VT) retinal crescent (**Mason and Slavi, 2020**). While some I-RGCs have been observed to project from the dorsocentral retina during embryonic stages, these are rapidly eliminated so-called 'transient' I-RGCs (**Soares and Mason, 2015**). Thus, in adulthood, C-RGCs are present throughout the retina while 'permanent' I-RGCs are confined to the VT crescent.

The zinc-finger TF *Zic2* is expressed in a subset of postmitotic RGCs in VT retina and is both necessary and sufficient for establishing their ipsilateral identity (**Herrera et al., 2003**); transient dorsolateral I-RGCs do not express *Zic2* (**Pak et al., 2004**). *Isl2* marks a subset of C-RGCs throughout the retina and appears to specify a contralateral identity in part by repressing Zic2 (**Pak et al., 2004**). These two TFs were expressed in a mutually exclusive fashion in RGC precursors between E13 and E16 (**Figure 7a**); *Zic2* was downregulated at later ages (**Figure 7—figure supplement 1a**). Furthermore, *Zic2* expression at E13 correlated with *Igfbp5* and *Zic1*, and anticorrelated with *Igf1* and *Fgf12*, consistent with recent reports (**Wang et al., 2016**; **Figure 7b**, **Figure 7—figure supplement 1b and c**). We scored each cell at E13 based on its expression of ipsilateral genes (Materials and methods), confirming that the expression of ipsilateral and contralateral gene signatures was anticorrelated (**Figure 7c**). Together, these observations support the idea that at E13 *Zic2+* cells represent I-RGCs and *Isl2+* cells represent some but not all C-RGCs.

Using WOT, we then identified the descendants of presumptive I-RGCs at later ages. We found that I-RGCs comprised 4.3% of adult RGCs, consistent with the range of 3–5% noted above. We queried these cells to identify the genes that distinguished putative I-RGCs and C-RGCs throughout the developmental time course. At a fold change of ≥1.5, we found 59 differentially expressed (DE) genes at E13 and 89 at E14 (**Figure 7e and f**). In addition to *Zic2, Igfbp5, Isl2,* and *Igf1*, which had been used to define I-RGCs and C-RGCs at E13, they included *Igfbpl1, Pou3f1,* and *Cntn2* enriched in I-RGCs, and *Lmo2, Pcsk1n,* and *Syt4* enriched in C-RGCs. The number of genes differentially expressed between

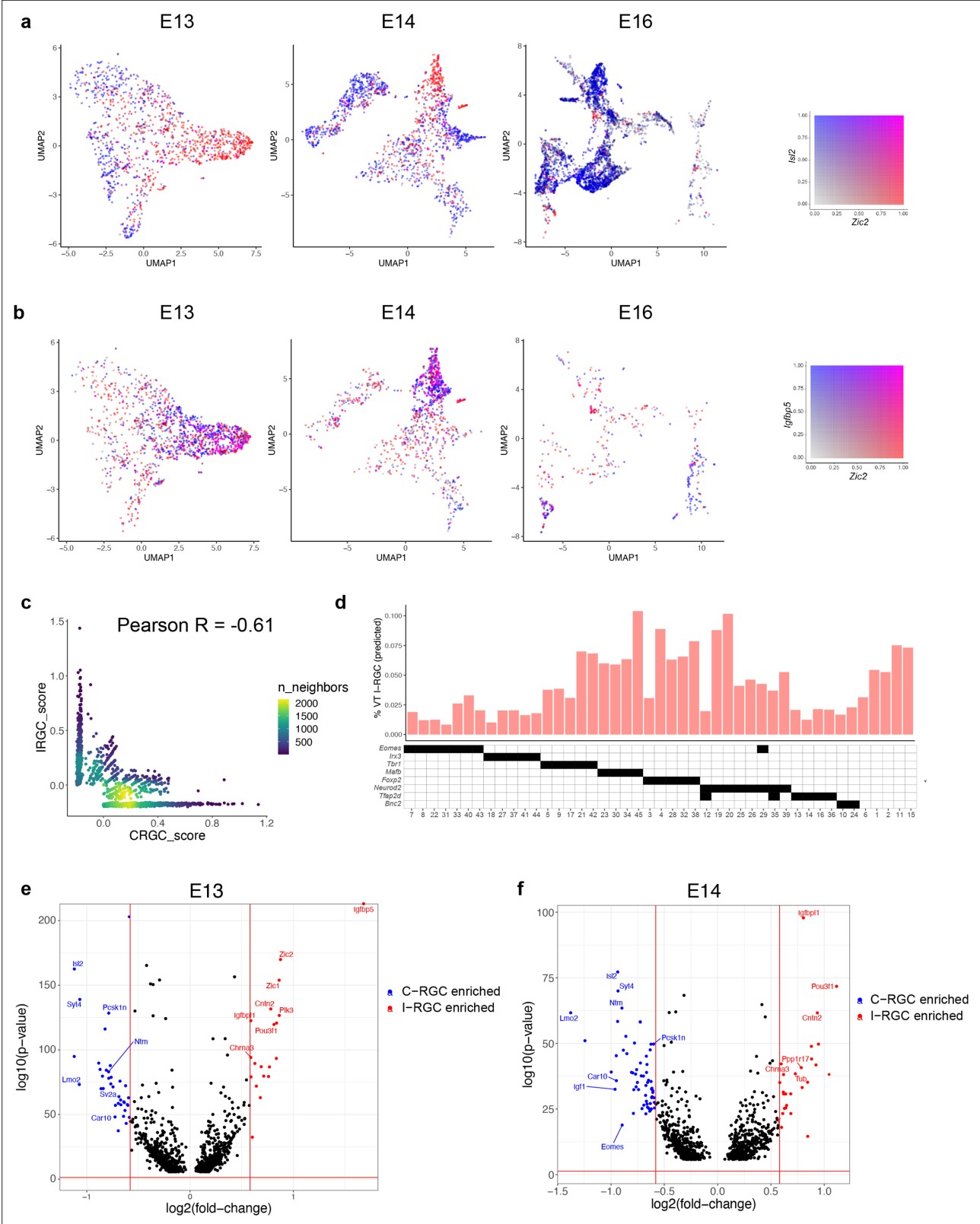

**Figure 7.** Transcriptomic identification of ipsilaterally projecting retinal ganglion cells (RGCs). (**a**) *Zic2*, an I-RGC marker, and *Isl2*, a C-RGC marker, are expressed in a mutually exclusive pattern at embryonic day (E)13 (left), E14 (middle), and E16 (right). *Zic2* is undetectable after E16 (*Figure 7—figure supplement 1a*). Cells are colored based on a bivariate color scale representing co-expression of two markers (colorbar, right). (**b**) *Zic2* and *Igfbp5*, two I-RGC markers, are co-expressed at E13 (left) and E14 (middle). Representation as in panel (**a**). (**c**) Scatter plots of gene signatures used to identify

*Figure 7 continued on next page*

*Figure 7 continued*

I-RGCs (y-axis) and C-RGCs (x-axis) at E13 are negatively correlated (Pearson $R$ = –0.61). Each dot corresponds to a cell, the color represents the number of cells located at a particular (x, y) location (see colorbar, right). (**d**) Barplot showing % of putative I-RGCs (y-axis) within each of the 45 adult RGC types, estimated by computing the descendants of E13 I-RGCs using Waddington optimal transport (WOT). RGC types are arranged along the x-axis based on their membership of transcription factor (TF) groups shown in ***Figure 6a*** (annotation matrix, bottom). (**e**) Volcano plot showing differentially expressed genes (MAST test) between predicted I-RGCs and C-RGCs at E13. The x- and the y-axes show the fold change and the p-value in log2- and log10- units, respectively. Dots represent genes, with red and blue dots highlighting I- and C-RGC-enriched genes, respectively, at fold change >1.5 and Bonferroni-corrected p-value<$5 \times 10^{-5}$. The two vertical bars correspond to a fold change of 1.5 in either direction. Select I-RGC- and C-RGC-enriched genes are labeled. (**f**) Same as panel (**e**), for E14.

The online version of this article includes the following figure supplement(s) for figure 7:

**Figure supplement 1.** Transcriptomic analysis of retinal ganglion cell (RGC) laterality.

I- and C-RGCs decreased after E14, with 20, 9, and 0 significant genes at E16, P0, and P5, respectively (***Figure 7—figure supplement 1d and e***), presumably reflecting the downregulation of axon guidance programs once retinorecipient targets have been reached (see Discussion).

We also asked which RGC types included I-RGCs. At E13, putative I-RGCs were highly enriched in 2 of 10 clusters, comprising 38–40% of clusters 2 and 9, 9–14% of clusters 3 and 5, and <2% of the other six clusters (***Figure 7d***). In adults, RGCs expressing *Tbr1*, *Mafb*, *Foxp2*, and *Neurod2* contained 3–4× more I-RGCs than RGCs expressing *Eomes*, *Irx3*, or *Tfap2d*. These results are consistent with previous observations that I-RGCs are morphologically and physiologically heterogenous but not uniformly distributed across types (***Hong et al., 2011***; ***Johnson et al., 2021***). Lastly, the WOT-predicted relationship between E13 precursor RGC clusters and I-RGC-rich or -poor terminal types was consistent with these patterns. The top six I-RGC-rich types (C4, C15, C19, C20, C38, and C45) derived 14 and 4% of their relative fate association from E13 clusters 2 and 9, while the top six I-RGC-poor types (C8, C14, C18, C22, C31, and C41) derived only 3.8 and 0.2% of their relative fate association from E13 clusters 2 and 9. Thus, E13 clusters 2 and 9 are preferred precursors of adult types that are relatively rich in I-RGCs.

## Discussion

The staggering diversity of its neurons underlies the computational power of the nervous system. Accordingly, a major quest in developmental neurobiology is to understand the mechanisms that diversify progenitors. A generally accepted way to deal with this diversity is to divide neurons into classes, and then subdivide classes into subclasses and subclasses into types (***Zeng and Sanes, 2017***). While much has been learned about how neural progenitors give rise to distinct neuronal classes, little is known about how classes diversify into subclasses and types.

Here, we used mouse RGCs to address this issue. We recently generated a molecular atlas that divided RGCs into ~45 distinct types based on their patterns of gene expression (***Tran et al., 2019***). We used this atlas here as a foundation to ask how these types are specified during development. We conclude that the earliest precursor RGCs are multipotential and exhibit continuous variation in transcriptomic identity, then diversify into definitive types by a gradual process of fate restriction. Interestingly, these features resemble those that have been discovered to control the generation of retinal cell classes from cycling progenitors (RPCs): multipotentiality, progressive restriction of fate, and stochastic rather than deterministic fate choice (see Introduction). We suggest that, at least in this case, similar strategies are used to generate cell classes from mitotically active progenitors and cell types from postmitotic precursors.

### Classes, subclasses, and types

Definitions of neuronal class, subclass, and type have been contentious (***Yuste et al., 2020***). In general, classes share general features of structure, function, molecular architecture, and location, whereas types comprise the smallest groups within classes that can be qualitatively distinguished from other groups based on these and other criteria. Subclasses lie in-between. For RGCs, class identity has been clear for a century, but inventories of subclasses and types have emerged only over the last few decades as high-throughput methods have been implemented for quantifying structural (primarily dendritic morphology), functional (responses to an array of visual stimuli), and molecular properties

(gene and transgene expression) of large numbers of RGCs. Fortunately, to the extent that they have been compared, there is excellent concordance among types defined by molecular, structural, and physiological criteria (*Bae et al., 2018*; *Goetz et al., 2021*; *Tran et al., 2019*; see http://www.rgc. types.org). Moreover, RGCs of a single type exhibit a regular spacing, called a mosaic arrangement, in that they tend to avoid other members of the same cell type, whereas their association with members of other types is random (*Kay et al., 2012*; *Keeley et al., 2020*; *Rockhill et al., 2000*). The molecular basis of this property is poorly understood, but it provides an additional criterion for defining a type. Thus, while no two RGCs are identical, and variation may be continuous in some other structures (*Cembrowski and Spruston, 2019*), there is strong reason to believe that RGC types are discrete.

## The adult RGC atlas

Developmental trajectories of cell types cannot be better than the adult types at which they are aimed. We have two reasons to believe that our adult RGC atlas (*Tran et al., 2019*) is accurate and complete.

First, the atlas is based on a detailed analysis of 35,699 cells and is therefore powered to detect types occurring at ~0.1% frequency (>40 cells per type; https://satijalab.org/howmanycells/). Results were stable over a variety of parameters (*Tran et al., 2019*). Moreover, in the course of studies on responses of RGCs to injury, we recently profiled an additional ~120,000 cells (A. Jacobi, N. Tran, W Yan, and J.R.S, in preparation), without identifying additional types.

Second, RGCs have now been classified by functional and structural properties, based on physiological responses to visual stimuli (*Baden et al., 2016*; *Goetz et al., 2021*) and serial section electron microscopy (*Bae et al., 2018*). The numbers of types defined in these ways (47 in *Bae et al., 2018*, 42 in *Goetz et al., 2021*, and >32 in *Baden et al., 2016*) match well to the 45–46 defined molecularly (*Tran et al., 2019*).

## Multipotentiality of precursor RGCs

The multipotentiality of dividing progenitor cells can be demonstrated by indelibly labeling a progenitor and then examining its progeny at a later stage. For mammals, this was initially done by infecting single cells with a recombinant retrovirus encoding a reporter gene that could be detected following multiple cell divisions (*Price et al., 1987*; *Sanes et al., 1986*; *Turner and Cepko, 1987*). More recently, it has become possible to greatly increase throughput by tracking scars or barcodes introduced by CRISPR/Cas9 (*Baron and van Oudenaarden, 2019*; *Espinosa-Medina et al., 2019*; *McKenna et al., 2016*). In sharp contrast, conclusively demonstrating that a single postmitotic cell is multipotential would require following a cell from an unspecified to a specified state, then turning back time, watching it again, and asking if it acquired the same mature identity. Since this is impossible, we used computational methods to draw tentative conclusions about the extent to which newly postmitotic RGCs are committed to mature into a particular type based on their transcriptomic profiles. Consequently, our results are based on inferred rather than experimentally determined lineages.

Our analysis proceeded in three steps. First, to ask whether RGCs were committed to a particular fate before or shortly after they were born, we assessed transcriptomic heterogeneity at a time when a large fraction was newly postmitotic (E13 and E14). We found that heterogeneity was present but limited: 10 transcriptomic clusters were distinguishable at E13 and 12 at E14. Thus, some heterogeneity is present in precursor RGCs, but far less than would be required to specify type identity before or immediately after their birth. A relevant issue is whether at these early stages the transcriptomic variation among cells could reflect variation in their stages of differentiation, perhaps as a consequence of different intervals between birthdate and sampling. Although we cannot exclude this possibility, inter-cluster variability of key RGC-specific genes (e.g., *Rbpms*) was no greater at early times than in adults (data not shown).

Second, we used a supervised classification approach to ask whether precursor RGC clusters mature into mutually exclusive sets of adult types. This model would imply an orderly, step-wise restriction of cell fates. However, our results indicate substantial overlap in the types derived from cells in different immature clusters. This result argues against a deterministic model of diversification and suggests that precursor RGCs are incompletely committed to a specific type for a substantial period after they are generated.

Third, we used optimal transport inference (WOT) to ask whether the multipotentiality observed at the level of groups was also a property of individual cells. This approach circumvents the limitation of the supervised classification approach, which compares similarity only at the level of clusters. WOT utilizes time-course scRNA-seq snapshots to infer fate associations between individual cells sampled at different time points, without reference to the clusters in which they reside (*Schiebinger et al., 2019*). While being consistent with supervised classification results at the cluster level, WOT indicated that the majority of individual RGCs were multipotential at E13 and E14. Of equal importance, immature RGCs were not totipotential: the average predicted potential ($P$) was 11.6 at E13, or ~25% of the maximum possible value of 45, and no RGCs had $P > 30$. We conclude that single multipotential immature RGCs are biased in favor of particular groups of adult RGC types.

## Progressive restriction of RGC fate

Further analysis provided insight into the structure of multipotentiality among RGCs. The adult RGC types associated with a precursor RGC were not a randomly chosen subset; rather some were more likely to arise from a common precursor state ('fate coupled') than others. This suggests a model in which RGC types arise via a progressive decoupling of fates within multipotential precursors. Decoupling is asynchronously, with different types emerging at different times. By modeling the temporal kinetics of fate decoupling, we were able to estimate a tentative specification time for each type – that is, the time at which precursors acquire a distinct transcriptomic identity. Analysis of transcriptomic changes that occur during this process, and the effects of visual experience on maturation, will be presented elsewhere (K.S., I.E.W., S.B. and J.R.S., in preparation).

Our conclusions about fate restriction are based on analyzing cell states defined by expression patterns of highly variable genes (HVGs) identified in the data (*Figure 2h*). An alternative and common view is that if a small set of genes is sufficient to define cell state, they should be the focus of analysis. We believe this is an incorrect argument based on confusion between genes that determine a cell type or state and those that define it. The only way a small number of genes, whatever their functional role, can exclusively define cell state is if they are expressed at very high levels. If this were the case, they would be detected in our data and drive the clustering.

## Fate-restricted RGC subclasses

For RGCs, class identity has been clear for a century, and type identity has been solidified during over the last few decades, but criteria for defining subclasses remain unclear. Tentative classifications have used molecular, physiological, and morphological criteria (*Sanes and Masland, 2015*; *Tran et al., 2019*). In general, these criteria correlate imperfectly with each other, a main exception being that ON and OFF RGCs (responding preferentially to increases and decreases in illumination, respectively) have dendrites that arborize in the inner and outer portions of the inner plexiform layer (*Famiglietti and Kolb, 1976*).

The pattern of fate couplings between RGC types at E13–14 provides an alternative way to define RGC subclasses – groups of RGC types that arise from the restriction of a common transcriptionally defined precursor state. We identified TFs selectively expressed within these subclasses. Our rationale was that among them would be fate determinants, an idea that could be tested by conventional genetic manipulations. Support for this idea is that there is already strong evidence that one such factor is a fate determinant in mouse: *Eomes* is selectively expressed by ipRGCs (and a few other types), and *Eomes* mutants fail to form ipRGCs although their retinas are normal in many respects (*Mao et al., 2014*). This encourages the hope that some of the other TFs in this set are also fate determinants. It will also be interesting to determine whether members of fate-restricted subclasses share structural or functional properties.

## Laterality

The TFs *Isl2* and *Zic2* are selective markers of embryonic RGCs that project contralaterally or ipsilaterally, respectively, and are critical determinants of this choice (*Herrera et al., 2003*; *Pak et al., 2004*). We found that their expression was largely nonoverlapping in RGCs at E13, and that they were co-expressed with previously reported markers of contralaterally and ipsilaterally projecting RGCs, respectively. Because few RGC axons reach the optic chiasm before E14, our results are consistent with genetic evidence that this differential expression is a cause rather than a consequence of the

divergent choices the axons make at the chiasm. Among many genes co-expressed with *Isl2* or *Zic2* may be others that play roles in this choice.

*Zic2* is downregulated later in embryogenesis, so we selected some RGCs as putative I-RGCs using genes known to be expressed in them (*Wang et al., 2016*), then used WOT to infer the RGC types to which they gave rise. Our analysis suggests that I-RGCs comprise many differentiated types, consistent with previous results (*Hong et al., 2011*; *Johnson et al., 2021*). Surprisingly, however, there were few if any genes differentially expressed between the putative mature I- and C-RGCs. Assuming that WOT results are valid – an assertion that can be tested directly in the future – this result suggests a model in which newborn RGCs are doubly specified – by laterality and type – but that once axonal choice has been made the laterality program is shut down.

## Beyond the retina

Generation of neuronal classes has been analyzed in many parts of the vertebrate nervous system, but we are aware of few reports on how classes diversify into types. A recent study addressed this issue for primary sensory neurons and reached the conclusion that newborn neurons in dorsal root ganglia are transcriptionally unspecialized and become type-restricted as development proceeds (*Sharma et al., 2020*). Similarly, both excitatory neuronal subclasses appear to diversify postmitotically in the mouse cerebral cortex (*Di Bella et al., 2021*; *Lodato and Arlotta, 2015*), and there is suggestive evidence that the same is true for interneuronal subclasses (*Wamsley and Fishell, 2017*). In all of these cases, it is attractive to speculate that diversification may occur by a process of fate decoupling in subpopulations of distinct multipotential precursors, akin to that documented here for RGCs. Our study provides a computational framework for investigating this issue.

# Materials and methods

**Key resources table**

| Reagent type (species) or resource | Designation | Source or reference | Identifiers | Additional information |
|---|---|---|---|---|
| Strain, strain background (*Mus musculus*) | C57BL/6 | Charles River or Jackson Labs | Cat#JAX000664; RRID: IMSR_JAX:000664 | |
| Antibody | Anti-Thy1/anti-CD90 (rat monoclonal) | Thermo Fisher Scientific | #17-0902-82 | 1:200 |
| Antibody | Anti-L1cam (rat monoclonal) | Miltenyi Biotec | #130-102-243 | 1:10 |
| Antibody | Anti-CD90 pre-conjugated (rat monoclonal) | Miltenyi Biotec | #130-049-101 | 1:200 |
| Antibody | Anti-RBPMS (guinea pig polyclonal) | PhosphoSolutions | #1832-RBPMS | 1:1000 |
| Antibody | Anti-KI67 (rabbit monoclonal) | Thermo Fisher Scientific | #MA5-14520 | 1:250 |
| Chemical compound, drug | Fluoro-Gel | Electron Microscopy Sciences | #17985 | |
| Commercial assay or kit | MACS Large Cell Columns | Miltenyi Biotec | #130-042-202 | |
| Sequence-based reagent | Drop-seq beads | ChemGenes Corporation | #Macosko201110 | |
| Commercial assay or kit | Papain dissociation system | Worthington | #LK003160 | |
| Commercial assay or kit | Chromium Single Cell 30Library and Gel Bead Kit v2, 10X Genomics 16rxns | 10X Genomics | #120237 | |
| Software, algorithm | Cell Ranger v2.6.0 | 10X Genomics | https://support.10xgenomics.com/single-cell-gene-expression/software/downloads/latest | |
| Software, algorithm | ImageJ (Fiji) version 2.1.0 | Fiji | https://imagej.net/Fiji | |
| Software, algorithm | R 3.6.2 | The R Foundation | https://www.r-project.org/ | |
| Software, algorithm | RStudio 1.3.1056 | RStudio | https://www.adobe.com | |

## Mice

All animal experiments were approved by the Institutional Animal Care and Use Committees (IACUC) at Harvard University. Mice were maintained in pathogen-free facilities under standard housing

conditions with continuous access to food and water. Animals used in this study include both males and females. A meta-analysis (not shown) did not show any systematic sex-related effects in either DE genes or cell-type proportions. For scRNA-seq and histology, we used C57Bl/6J (JAX #000664). Embryonic and early postnatal C57Bl/6J mice were acquired either from Jackson Laboratories (JAX) from time-mated female mice or time-mated in-house. For timed-matings, a male was placed with a female overnight and removed the following morning (with the corresponding time recorded as E0.5).

### Cell preparation

RGCs were enriched from dissociated retinal cells as previously described with minor modifications (*Tran et al., 2019*). All solutions were prepared using Ames' Medium with L-glutamine and sodium bicarbonate (equilibriated with 95% $O_2$/5% $CO_2$), and all spin steps were done at 450 × $g$ for 8 min. Retinas were dissected out in their entirety immediately after enucleation and digested in ~80 U of papain at 37°C, with the exception of some E13 and E14 eyes that were digested whole, followed by manual trituration in ovomucoid solution. Clumps were removed using a 40 μm cell strainer and the cell suspension was spun down and resuspended in Ames + 4% BSA at a concentration of 10 million cells per 100 μl. Cells from E13, E14, E16, and P0 were incubated for 15 min at room temperature with antibodies to Thy1 (also known as CD90) and L1CAM pre-conjugated to the fluorophores APC (Thermo Fisher Scientific #17-0902-82) and PE (Miltenyi Biotec #130-102-243), respectively. Cells were washed with 6 ml of Ames + 4% BSA, spun down and resuspended at a concentration of ~7 million cells/ml, and calcein blue was added to label metabolically active cells.

Viable Thy1- or L1CAM-positive cells were sorted using a MoFlo Astrios sorter into ~100 μl of AMES + 4% BSA. Sorted cells were spun down a final time and resuspended in PBS + 0.1% BSA at a concentration of 500–2000 cells/μl. P5 RGCs were enriched using only CD90, with either magnetic-activated cell sorting (MACS) using large cell columns and CD90 pre-conjugated to microbeads (#130-042-202 and #130-049-101, Miltenyi Biotec) or fluorescence-activated cell sorting with CD90 pre-conjugated to PE/Cy7 (Thermo Fisher Scientific #25-0902-81), or both.

### Droplet-based single-cell RNA-seq

#### Statement on replicates

We profiled immature RGCs using scRNA-seq at five developmental time points: E13, E14, E16, P0, and P5. At each age, data was collected from four replicate experiments. Experiments at E13, E14, E16, and P0 involved two biological replicates (distinct mice). Each of these biological replicates was further subdivided into two equal pools, and the cells were subjected to two different enrichment methods (anti-Thy1 and anti-L1cam). Thus, each of these time points consisted of four replicate experiments. RGC enrichment at P5 exclusively utilized anti-Thy1, but four biological replicate experiments were performed. One of these was profiled using 10X, and three of these were profiled using Drop-seq.

#### Drop-seq

A subset of P5 RGC dataset was collected using Drop-seq (*Macosko et al., 2015*), performed largely as described previously (*Shekhar et al., 2016*). Briefly, cells were diluted to an estimated final droplet occupancy of 0.05, and co-encapsulated in droplets with barcoded beads, which were diluted to an estimated final droplet occupancy of 0.06. The beads were purchased from ChemGenes Corporation, Wilmington, MA (# Macosko201110). Individual droplet aliquots of 2 ml of aqueous volume (1 ml each of cells and beads) were broken by perfluorooctanol, following which beads were harvested, and hybridized RNA was reverse transcribed. Populations of 2000 beads (~100 cells) were separately amplified for 14 cycles of PCR (primers, chemistry, and cycle conditions identical to those previously described) and pairs of PCR products were co-purified by the addition of 0.6x AMPure XP beads (Agencourt). Fifteen experimental replicates were sequenced in total from five biological replicates using an Illumina NextSeq 500. Read 1 was 20 bp; read 2 (paired-end) was 60 bp.

#### 10X Genomics

Single-cell libraries were prepared using the single-cell gene expression 3′ kit on the Chromium platform (10X Genomics, Pleasanton, CA) following the manufacturer's protocol. As our datasets were collected over a long period of time, we used a combination of v1 (a single channel of P5 RGCs) and

v2 (E13, E14, E16, P0). Briefly, single cells were partitioned into Gel beads in EMulsion (GEMs) in the 10X Chromium instrument followed by cell lysis and barcoded reverse transcription of RNA, amplification, enzymatic fragmentation, 5′ adaptor attachment, and sample indexing. On average, approximately 8000–12,000 single cells were loaded on each channel and approximately 3000–7000 cells were recovered. Libraries were sequenced on the Illumina HiSeq 2500 platforms at the Broad Institute (paired-end reads: read 1, 26 bases; read 2, 98 bases).

## Power analysis

An important question in all single-cell experiments is that of the number of cells to profile. A widely used approach is the power analysis tool published by the Satija lab (https://satijalab.org/howmany-cells/). Fortunately, in this study we were also guided by our previous study of adult RGCs, where we had knowledge of the frequency distribution of adult RGC types, with the rarest type being approximately 0.2% (*Tran et al., 2019*). In that study, we also found that when classification is performed in a supervised fashion based on an existing atlas, approximately ~8000 RGCs were sufficient to recover the accurate relative frequency distribution of 45 RGC types. We therefore aimed to profile ~8000 cell at each time point as our analysis involved mapping immature RGCs to the adult atlas. With the exception of E13, all time points contain 1.5–2× more cells than this target value.

## Histology

### Tissue fixation

Adult (P56) mice were intracardially perfused with 2–5 ml of PBS followed by 15 ml of 4% PFA, followed by additional fixation of eyes for 15 min in 4% PFA. P0 and P5 mice were not perfused, rather eyes were fixed in 4% PFA for 30 min. At E13, 14, and 16, embryos were fixed whole for 30 min in 4% PFA, following which eyes were removed. Following fixation, eyes from all time points were transferred to PBS and stored at 4°C until subsequent use.

### Sectioning

Cross sections for immunohistochemistry (IHC) were generated using a Leica CM1850 cryostat. For some early developmental time points, eyes were kept whole for IHC, otherwise retinas were either (1) dissected out in their entirety from eyes or (2) the cornea, iris, and lens were removed, leaving the sclera and retina intact. Tissues were sunk in 30% sucrose overnight at 4°C, embedded in tissue freezing medium, and cryo-sectioned into 25 mm slices. Slides with tissue sections were air-dried for ~3 hr and stored at –80°C until staining.

### Immunohistochemistry

All IHC solutions were made up in PBS + 0.3% Triton-X, and all incubation steps were carried out in a humidified chamber. Following a 1 hr protein block in 5% normal donkey serum at room temperature, slides were incubated overnight at 4°C with primary antibodies, washed twice for 5 min each in PBS, incubated for 2 hr at room temperature with secondary antibodies conjugated to various fluorophores (1:1000, Jackson Immunological Research) and Hoechst (1:10,000, Life Technologies), and washed again twice for 5 min each in PBS before coverslipping with Fluoro-Gel (#17985, Electron Microscopy Sciences). Primary antibodies used include guinea pig anti-RBPMS (1:1000, #1832-RBPMS, Phospho-Solutions), rabbit anti-KI67 (1:250, #MA5-14520, Thermo Fisher Scientific), and rat anti-L1CAM (1:10, #130-102-243, Miltenyi Biotec).

### Imaging

All images were acquired using an Olympus Fluoview 1000 scanning laser confocal microscope, with a ×20 oil immersion objective and ×2 optical zoom. Optical slices were taken at 1 μm steps. Fiji was used to pseudocolor each channel and generate a maximum projection from image stacks. Brightness and contrast were adjusted in Adobe Photoshop.

## Alignment and quantification of gene expression in single cells

All single-cell libraries were aligned to the UCSC mm10 transcriptomic reference (*Mus musculus*), and gene expression matrices were quantified using standard protocols described previously. For

the single-cell libraries generated using the 10X platform (E13, E14, E16, P0, and P5), these steps were performed using cellranger v2.1.0 (10X Genomics). For the single-cell libraries generated using Drop-seq (P5), we used Drop-seq tools (v1.12; *Macosko et al., 2015*), following the procedures described earlier (*Shekhar et al., 2016*). Alignment and quantification was done for each sample library separately to generate a genes × cells expression matrix of transcript counts. These matrices were column-concatenated for further analysis.

We retained cells that expressed at least 700 genes, resulting in 98,452 cells. We also removed genes expressed in fewer than 10 cells. The resulting $M$ genes × $N$ cells matrix of UMI counts $C_{mn}$ was normalized along each column (cell) to sum to 8340, the median of the column sums resulting in a normalized matrix $X_{mn}$. This was followed by the transformation $X_{mn} \leftarrow \log(X_{mn} + 1)$.

## Overview of clustering analysis

The following procedure was adopted to perform batch correction, dimensionality reduction, and clustering throughout the article. The procedure was first applied on the entire dataset to separate RGCs from other cell classes, and then to RGCs at each age to identify transcriptomically distinct groups.

1. *Identification of HVGs:* We used the Gamma-Poisson framework described previously to identify HVGs (*Pandey et al., 2018*). Briefly, we compute for each gene the mean ($\mu_m$) and the coefficient of variation ($CV_m$) for the UMI counts $C_{mn}$,

   $\mu_m = \frac{1}{N} \sum_n C_{mn}$
   $\sigma_m^2 = \frac{1}{N} \sum_n (C_{mn} - \mu_m)^2$
   $CV_m = \frac{\sigma_m}{\mu_m}$

   For a given $\mu_m$, the Gamma-Poisson model predicts a 'null' coefficient of variation ($CV_m^{null}$) arising from a combination of Poisson 'shot' noise and large variations in library size, assumed to be due to technical reasons,

   $CV_m^{null} = \frac{1}{\mu_m} + \frac{1}{\alpha}$

   Here, $\alpha$ is the shape parameter of a Gamma-distribution fit to the distribution of normalized library sizes $T_n$ (using the R package `MASS`),

   $T_n = \frac{\sum_m C_{mn}}{\sum_{m,n} C_{mn}}$

   In practice, $CV_m^{null}$ serves as a tight lower bound for empirically observed values of $CV_m$ across the full range of $\mu_m$. This enables us to compute for each gene $m$, a deviation score $d_m = \log \frac{CV_m}{CV_m^{null}}$, quantifying the extent to which its observed coefficient of variation exceeds the predicted null model. HVGs are selected if they satisfy $d_m > Mean(d_m) + 0.8\, Std(d_m)$.

2. *Batch correction and dimensionality reduction*: We subsetted the rows of the expression matrix $X_{mn}$ to the HVGs identified in Step 1. As our data comprised cells sampled at different developmental ages as well as multiple biological replicates within each age, we used Liger, a nonnegative matrix factorization technique, to embed the data in a reduced dimensional latent space of shared factors (*Welch et al., 2019*). Liger computes a factorized representation for each matrix that separates 'shared' and 'dataset-specific' gene expression modules (factors). We use Liger's normalized $H$ factor loadings for cells to build a nearest-neighbor graph and define clusters.

   As in any matrix factorization technique, Liger requires the user to choose $k$, the dimensionality of the latent space. To find $k$, we use a Random Matrix Theory approach (outlined in *Peng et al., 2019*). Briefly, $k$ is estimated as the number of eigenvalues of the sample gene–gene correlation matrix that exceed the 99th percentile of the distribution of the largest eigenvalue of a random Hermitian matrix of the same dimensions. This is given by the Tracy–Widom distribution. For these calculations, we used the R package RMTstat.

3. *Clustering and 2D visualization*: To cluster cells based on transcriptomic similarity, we first built a nearest-neighbor graph on the cells based on their normalized $H$ factor coordinates computed using Liger. The number of nearest neighbors was chosen to be 30. The edges were weighted based on the Jaccard overlap metric, and graph clustering was performed using the Louvain algorithm, as described previously (*Shekhar et al., 2016*). The normalized $H$ factor coordinates were also used as input to project cells on to a nonlinear 2D space using the Uniform Manifold Approximation and Projection algorithm (UMAP; *Becht et al., 2018*). Graph construction, clustering, and the UMAP projection were performed using the R packages FNN, igraph, and umap, respectively.

We began by clustering the full dataset combining all ages using the procedure outlined above. We identified groups of clusters corresponding to cell classes, which included RGCs (*Rbpms, Slc17a6, Sncg, Nefl*), microglia (*P2ry12, C1qa-c, Tmem119*), photoreceptors (*Otx2, Gngt2, Gnb3*), amacrine cells (*Tfap2a, Tfap2b, Onecut2*), anterior segment cells (*Mgp, Col3a1,Igfbp7*), cycling progenitors (*Ccnd1, Fgf15, Hes5*), and neurogenic progenitors (*Hes6, Ascl1, Neurog2*). Deeper annotation (e.g., of RGC type) was not done at this stage. No other cellular classes were identified. Three clusters comprising fewer than 1.2% of the cells expressed markers of more than one class. These were flagged as doublets and removed from further analyses.

## Defining RGC precursor heterogeneity at each age

RGC precursors at each age were separately analyzed following the clustering pipeline outlined previously. When implementing Liger, each biological replicate was regarded as a separate batch. The nominal clusters identified by the Louvain algorithm were refined as follows:

1. *Removing contaminants*: Clusters were flagged for further examination if they did not exclusively express RGC-specific markers (e.g., *Rbpms, Slc17a6, Sncg, Nefl*). These clusters were small (typically <1–2% of cells) and in all cases expressed non-RGC markers (e.g., *P2ry12* or *Tfap2b*). These cells, which likely reflect trace contaminants, were discarded from further analysis.
2. *Merging proximal clusters*: Transcriptomic relationships between nominated clusters were visualized on a dendrogram computed using hierarchical clustering, as noted above. Neighboring clusters on the dendrogram, which were leaves in a terminal branch, were assessed for differential expression using the MAST DE test (*Finak et al., 2015*). A gene *g* was regarded as significantly DE between clusters $\mathcal{C}_a$ and $\mathcal{C}_b$ if it satisfied $\left|\log FC_g\left(\mathcal{C}_a, \mathcal{C}_b\right)\right| > 0.5$ and MAST p-value was less than $10^{-5}$ (false discovery rate [FDR] corrected), where

$$\log FC_g\left(\mathcal{C}_a, \mathcal{C}_b\right) = \ln\left(\frac{|\mathcal{C}_b| \sum_{n \in \mathcal{C}_a} X_{gn}}{|\mathcal{C}_a| \sum_{n \in \mathcal{C}_b} X_{gn}}\right)$$

is defined to be the log-fold change in expression. Clusters that showed fewer than 10 significant DE genes were merged. In this manner, we identified 10 RGC clusters at E13, 12 at E14, 19 at E16, 27 at P0, and 38 at P5. Using MAST, we identified DE genes that distinguished each cluster against the rest at any given age.

## Quantifying RGC diversity at different ages

We quantified the molecular diversity of RGCs based on clusters at each stage using three measures of population diversity – the Rao index (*Figure 2*), the Shannon index, and the Simpson index (*Figure 2—figure supplement 1*). For *N* clusters with relative frequencies $p_1, p_2, \ldots, p_N$, these indices are defined as follows:

1. Let $d_{ij}$ be a distance measure between clusters *i* and *j* ($0 \leq d_{ij} \leq 1$). The *Rao index* is defined as We used varying number of genes (≈1200–3000) to calculate $d_{ij}$. The computed Rao index was insensitive to these variations.
2. The *Shannon index* is defined as
   $H = -\sum_i p_i \log p_i$
3. The *Simpson index* is defined as
   $S = \sum_i p_i^2$
4. While the Rao and Shannon indices increase with diversity, the Simpson index decreases with diversity.

## Analysis of cluster distinctiveness

We quantified the mutual separation of clusters at each age using two approaches:

1. *Multi-class classification*: We trained a multi-class classifier (R package `xgboost`) at each age on 50% of the cells using their cluster IDs. The remaining 50% of the cells were used to test the learned classifier and estimate a classification error per cluster, which were averaged at each age. As clusters become better separated, the average classification error decreases.
2. *Relative positions in PC space*: At each age, the top 20 principal component analysis (PCA) coordinates were first standardized by z-scoring. For each cluster *C* at a given age, we computed two quantities:

a. $r_C$ , the median of Euclidean distances of each cell from the cluster centroid in the standardized PCA coordinates.

b. $d_C$ , the median of Euclidean distances of each cell from the centroid of the nearest external cluster.

For a cluster C, a low of value $r_C/d_C$ indicates a higher degree of separatedness. Averaging this metric across all the clusters at a given age quantifies the degree to which clusters are separated in the UMAP representation.

## Relating clusters across ages using supervised classification

### Analysis overview

To distinguish between 'specified' and 'nonspecified' modes of diversification (*Figure 3*), we first used a supervised classification approach to associate immature RGC clusters at young ages (tests) to cluster IDs determined at older ages (references). We used XGBoost, a decision-tree-based ensemble learning algorithm (*Chen and Guestrin, 2016*), to train multi-class classifiers on reference clusters and used these to assign labels to individual test RGCs.

Two kinds of references were used: (1) classifiers trained on the adult (P56) clusters were used to assign immature RGCs at each of the five developmental ages (five separate analyses) to adult labels. (2) Classifiers trained on E14, E16, P0, and P5 clusters were used to assign E13, E14, E16, and P0 RGCs to labels corresponding to the previous age, respectively (four separate analyses). The correspondence between classifier assigned labels and cluster IDs of test RGCs was visualized using confusion matrices (e.g., *Figure 3d–h*) and quantified using two metrics – the ARI and NCE metrics, described below.

### Classification overview

To describe our classification analysis, we introduce some notation to facilitate a description in general terms. Let $\mathcal{A}^R$ and $\mathcal{A}^T$ denote the reference and the test atlases for the purpose of supervised classification. The number of cells (i.e., RGCs) contained in the reference and test atlases is denoted $|\mathcal{A}^R|$ and $|\mathcal{A}^T|$, respectively. $\mathcal{A}^R$ and $\mathcal{A}^T$ could correspond to any pair of ages as described above. Without loss of generality, let us assume that $\mathcal{A}^R$ contains $r$ transcriptomic clusters denoted $\left\{ \mathcal{C}_1^R, \mathcal{C}_2^R, \ldots \mathcal{C}_r^R \right\}$. Similarly, $\mathcal{A}^T$ is assumed to contain $t$ transcriptomically defined clusters denoted $\left\{ \mathcal{C}_1^T, \mathcal{C}_2^T, \ldots \mathcal{C}_t^T \right\}$.

Each cell in our dataset is the member of a particular atlas and is assigned to a single cluster within the atlas based on its transcriptome. The transcriptome of each cell is a vector (denoted using lowercase boldface symbols, e.g., $\boldsymbol{u}$ or $\boldsymbol{v}$) with the number of elements equal to the number of HVGs (the features used for classification). Let $cluster(\boldsymbol{u})$ denote the transcriptionally assigned of cell $\boldsymbol{u}$. For example, the following statement,

$$\boldsymbol{u} \in \mathcal{A}^T, \ \ cluster(\boldsymbol{u}) = \mathcal{C}_k^T ,$$

translates to 'Cell $\boldsymbol{u}$ in atlas $\mathcal{A}^T$ is a member of cluster $\mathcal{C}_k^T$ .' Our goal is to *assign* each cell $\boldsymbol{u} \in \mathcal{A}^T$ a second ID $cluster'(\boldsymbol{u})$ based on its transcriptomic correspondence to the reference atlas $\mathcal{A}^R$ . We perform this via an XGBoost classifier trained on $\mathcal{A}^R$ and applied it to every cell in $\mathcal{A}^T$ , allowing us to infer transcriptomic correspondences between the two sets of clusters. The main steps are as follows:

- The expression matrices in $\mathcal{A}^R$ and $\mathcal{A}^T$ are z-scored along each feature. The initial set of features are chosen as the common HVGs in the two atlases. Parameters are adjusted to select the common top ~2000–3000 HVGs.
- Classifiers $Class_0^R$ and $Class_0^T$ are trained on $\mathcal{A}^R$ and $\mathcal{A}^T$ independently. For training, we randomly sample 60% of cells in each cluster up to a maximum of 300 cells. The remaining 'held-out' cells are used for validation. We ran the training routine for XGBoost with the following parameter specification (see https://xgboost.readthedocs.io/en/latest/parameter.html):

```
xgb_params <- list("objective" = "multi:softprob",
"eval_metric" = "mlogloss",
"num_class"= nClusters,
"eta" = 0.2,"max_depth" = 6, subsample = 0.6)
```

- When applied to a cell vector $\boldsymbol{u}$, the classifier $Class_0^R$ (or $Class_0^T$) returns a vector of $\boldsymbol{p} = (p_1, p_2, \ldots)$ of length $r$ (or $t$) with entries representing probability values of predicted cluster memberships in the corresponding atlas. We use these values to compute the 'softmax' assignment of $\boldsymbol{u}$, so that $cluster'(\boldsymbol{u}) = argmax_i \, p_i$.
- Post training, $Class_0^R$ and $Class_0^T$ are evaluated on the respective validation sets. Using the predicted cluster assignments of the 'held out' cells, we compute for each cluster in $\mathcal{A}^R$ and $\mathcal{A}^T$ the error rate, defined as the fraction of held-out cells that were misclassified. If the error rate for any cluster was higher than 10%, the classifier is retrained by artificially upsampling cells from the high error rate clusters. In the final classifiers, the cluster-specific error rates were typically 1–4%, and in no case exceeded 10%.
- The top 500 discriminatory features (genes) are identified based on average information gain (using the function `xgb.importance`) for each of $Class_0^R$ and $Class_0^T$. These gene sets are denoted $\mathcal{G}^R$ and $\mathcal{G}^T$, respectively.
- The common features $\mathcal{G} = \mathcal{G}^R \cap \mathcal{G}^T$ are used to train a third classifier $Class^R$ on the reference atlas $\mathcal{A}^R$. This ensures that inferred transcriptomic correspondences are based on 'core' gene expression programs that are conserved at both time points rather than temporally regulated genes.
- Finally, $Class^R$ is applied to each cell $\boldsymbol{u} \in \mathcal{A}^T$ to generate predicted labels $cluster'(\boldsymbol{u})$. Global transcriptional correspondence was visualized using confusion matrices between cluster IDs $cluster(\boldsymbol{u}) \in \left\{ \mathcal{C}_1^T, \mathcal{C}_2^T, \ldots \mathcal{C}_t^T \right\}$ and reference assignments $cluster'(\boldsymbol{u}) \in \left\{ \mathcal{C}_1^R, \mathcal{C}_2^R, \ldots \mathcal{C}_r^R \right\}$, and their correspondence was quantified using the metrics described below.

## Quantifying cluster correspondence using global and local metrics

Let $N_{ij}$ denote the number of cells in $\mathcal{A}^T$ that are part of transcriptomic cluster $\mathcal{C}_j^T$ and are assigned by $Class^R$ to reference cluster $\mathcal{C}_i^R$. Thus,

$$N_{ij} = \# \left\{ cluster'(\boldsymbol{u}) = C_i^R, \, cluster(\boldsymbol{u}) = C_j^T \, \forall \, \boldsymbol{u} \in \mathcal{A}^T \right\}$$

$N_{ij}$ defines a contingency table, whose marginal sums are defined as

$$a_i = \sum_{j=1}^{t} N_{ij}$$

$$b_j = \sum_{i=1}^{r} N_{ij}$$

Let $N = \sum_{i,j} N_{ij} = \left| \mathcal{A}^T \right|$, the number of cells in $\mathcal{A}^T$. Then, the ARI corresponding to the assignments can be evaluated using the following equation:

$$ARI = \frac{\sum_{ij} \binom{N_{ij}}{2} - \left[ \sum_i \binom{a_i}{2} \sum_j \binom{b_j}{2} \right] / \binom{N}{2}}{\frac{1}{2} \left[ \sum_i \binom{a_i}{2} + \sum_j \binom{b_j}{2} \right] - \left[ \sum_i \binom{a_i}{2} \sum_j \binom{b_j}{2} \right] / \binom{N}{2}}$$

The *ARI* ranges from 0 and 1, with extremes corresponding to random association and 1:1 correspondences between $\mathcal{A}^R$ and $\mathcal{A}^T$, respectively. The ARI can technically also take on negative values for certain scenarios, but these are not observed in our data.

As an alternative, we also used the NCE, an information-theoretic measure. The NCE quantifies the extent to which knowledge of the value of $cluster'(\boldsymbol{u})$ reduces the uncertainty (measured in information bits) about the value of $cluster(\boldsymbol{u})$ for $\boldsymbol{u} \in \mathcal{A}^T$.

We introduce probability weights $q_{ij}$ and the corresponding marginals $q_{i,.}$ and $q_{.j}$ as follows:

$$q_{ij} = \frac{N_{ij}}{N}$$

$$q_{i,.} = \frac{a_i}{N}$$

$$q_{.j} = \frac{b_j}{N}$$

The conditional entropy (CE) is then given by the expression:

$$H\left(cluster\left(\boldsymbol{u}\right)|cluster'\left(\boldsymbol{u}\right)\right) = -\sum_{ij} q_{ij}\log\frac{q_{ij}}{q_{i.}}$$

Note that CE is asymmetric, that is, $H\left(cluster\left(\boldsymbol{u}\right)|cluster'\left(\boldsymbol{u}\right)\right) \neq H\left(cluster'\left(\boldsymbol{u}\right) cluster\left(\boldsymbol{u}\right)\right)$. One notes that $H = 0$ if for each cluster $i \in \{1, \ldots, r\}$, $q_{ij} = \delta_{i,k_i}$ for a single cluster $k_i$, where $\delta_{ij}$ is the Kronecker delta defined as

$$\delta_{ij} = 1, \; if \; i = j$$
$$\delta_{ij} = 0, \; if \; i \neq j$$

Finally, NCE is defined as

$$NCE = \frac{H(cluster(\boldsymbol{u})|cluster'(\boldsymbol{u}))}{(cluster(\boldsymbol{u}))}$$

where $H\left(cluster\left(\boldsymbol{u}\right)\right) = -\sum_j q_{.j}\log q_{.j}$ is the Shannon entropy. Due to this normalization, *NCE* values range from 0 to 1, with extremes corresponding to fully specific mapping or random association, respectively, between $\mathcal{A}^R$ and $\mathcal{A}^T$. *ARI* and *NCE* are inversely related. Unlike *ARI*, however, *NCE* is able to detect specificity in both many:1 and 1:1 mappings. ARI returns a value lower than 1 for specific mappings if the number of clusters in $\mathcal{A}^R$ and $\mathcal{A}^T$ is not equal.

*ARI* and *NCE* quantify global correspondences between $\mathcal{A}^R$ and $\mathcal{A}^T$. We also computed a local metric, the OF that quantified whether individual reference labels $\mathcal{C}_i^R$ were distributed in a 'localized' or 'diffuse' manner between test clusters$\in \left\{\mathcal{C}_1^T, \mathcal{C}_2^T, \ldots \mathcal{C}_t^T\right\}$.

$$OF\left(\mathcal{C}_i^R\right) = \frac{1}{t}\left[\frac{1}{\sum_j\left(\frac{q_{ij}}{q_{i.}}\right)^2}\right]$$

Note that the term $\frac{q_{ij}}{q_{i.}}$ is simply the fraction of the total test cells belonging to test cluster $\mathcal{C}_j^T$ that are assigned to reference cluster $\mathcal{C}_i^R$ by the classifier. Defined this way, the term in the square brackets computes an occupation number that ranges from 1 to $t$ and can be interpreted as the number of test clusters that are specifically associated with $\mathcal{C}_i^R$. Division by $t$ the number of test clusters, therefore, converts this number into a fraction.

## Overview of WOT analysis

To identify fate relationships among maturing RGCs, we used WOT (*Schiebinger et al., 2019*), a recently developed framework that is rooted in optimal transport theory (*Villani, 2009*). WOT does not rely on clustering, and therefore is able to identify ancestor–descendant relationships between any pair of temporally separated RGCs in our data.

At its heart, WOT models cellular transcriptomes $\boldsymbol{u}$ measured at a given age $t$ as a probability distribution in gene expression space $P_t(\boldsymbol{u})$ (note that $\boldsymbol{u}$ may represent the original gene expression space or a reduced dimensional embedding estimated via PCA or diffusion maps). This probability distribution evolves with time as cells differentiate and mature. Different temporal measurements collected at times $\ldots, t_{i-1}, t_i, t_{i+1}, \ldots$ represent temporal snapshots of the corresponding cell distributions $\ldots, P_{t_{i-1}}, P_{t_i}, P_{t_{i+1}}, \ldots$. Unfortunately, as each cell can only be measured once, the measurement at different times is from different cells. Therefore, for a particular cell $\boldsymbol{u}$ at time $t_i$, it is not clear which cell(s) at time $t_{i-1}$ are likely to be its ancestor(s) and which cell(s) at time $t_{i+1}$ are likely to be descendant(s). It is this problem that WOT addresses.

Briefly, for a given pair of consecutive transcriptomic snapshots $P_{t_i}(\boldsymbol{u})$ and $P_{t_{i+1}}(\boldsymbol{v})$, we wish to estimate the joint distribution $\boldsymbol{\Pi}_{t_i,t_{i+1}}\left(\boldsymbol{u},\boldsymbol{v}\right)$, representing the probability that a cell having an expression vector $\boldsymbol{u}$ at time $t_i$ transitions to a cell with an expression vector $\boldsymbol{v}$ at time $t_{i+1}$. $\boldsymbol{\Pi}_{t_i,t_{i+1}}\left(\boldsymbol{u},\boldsymbol{v}\right)$ is also called the temporal coupling, which, owing to the destructive nature of scRNA-seq assays, is not directly observable. Under the assumption that cells move short distances in transcriptomic space when $\Delta t_i = t_{i+1} - t_i$ is 'reasonably close,' WOT estimates $\boldsymbol{\Pi}_{t_i,t_{i+1}}\left(\boldsymbol{u},\boldsymbol{v}\right)$ as the solution to the following convex optimization problem:

$$\hat{\Pi}_{t_i,t_{i+1}} = argmin_\Pi \sum_{\boldsymbol{u}\in\mathcal{A}^{t_i}} \sum_{\boldsymbol{v}\in\mathcal{A}^{t_{i+1}}} c\left(\boldsymbol{u},\boldsymbol{v}\right) \Pi\left(\boldsymbol{u},\boldsymbol{v}\right) - \epsilon \int\int \Pi\left(\boldsymbol{u},\boldsymbol{v}\right) \log \Pi\left(\boldsymbol{u},\boldsymbol{v}\right) d\boldsymbol{u}d\boldsymbol{v}$$

$$+\lambda_1 KL\left[\sum_{\boldsymbol{u}\in\mathcal{A}^{t_i}} \Pi\left(\boldsymbol{u},\boldsymbol{v}\right) \| d\hat{P}_{t_{i+1}}\left(\boldsymbol{v}\right)\right] + \lambda_1 KL\left[\sum_{\boldsymbol{v}\in\mathcal{A}^{t_{i+1}}} \Pi\left(\boldsymbol{u},\boldsymbol{v}\right) \| d\hat{Q}_{t_i}\left(\boldsymbol{u}\right)\right]$$

In the above equation,

- $\hat{P}_{t_{i+1}}\left(\boldsymbol{v}\right)$ is an empirical distribution constructed from $\mathcal{A}^{t_{i+1}}$, which denotes the scRNA-seq atlas at $t_{i+1}$,
  - $\hat{P}_{t_{i+1}}\left(\boldsymbol{v}\right) = \frac{1}{|\mathcal{A}^{t_{i+1}}|} \sum_{\boldsymbol{x}_i\in\mathcal{A}^{t_{i+1}}} \delta\left(\boldsymbol{v}-\boldsymbol{x_i}\right)$
- where $\delta(\boldsymbol{v}-\boldsymbol{x})$ denotes the Dirac delta function, a probability distribution placing all its mass at the location $\boldsymbol{x}$.
- $\hat{Q}_{t_i}\left(\boldsymbol{u}\right)$ is the cell distribution at $t_i$ rescaled by the relative growth rate to account for cell division/death,
  - $\hat{Q}_{t_i}\left(\boldsymbol{u}\right) = \hat{P}_{t_i}\left(\boldsymbol{u}\right) \frac{g(\boldsymbol{u})^{t_{i+1}-t_i}}{\int g(\boldsymbol{u})^{t_{i+1}-t_i} d\hat{P}_{t_i}}$
- Here, $g\left(\boldsymbol{u}\right)$ represents the relative growth rate of cell $\boldsymbol{u}$ in the time interval $\left(t_i, t_{i+1}\right)$ and is estimated within the framework of unbalanced optimal transport (*Chizat et al., 2018*). For more details, we refer the reader to the supplementary information of *Schiebinger et al., 2019*.
- $c\left(\boldsymbol{u},\boldsymbol{v}\right)$ is a cost function defined as the Euclidean distance $|\boldsymbol{u}-\boldsymbol{v}|^2$. The first term of the objective function minimizes the cost function weighted by the temporal couplings, which may be interpreted as the transport distance between the distributions $\hat{P}_{t_i}$ and $\hat{P}_{t_{i+1}}$ (also known as the Wasserstein distance).
- The second term on the RHS represents entropic regularization, and $\epsilon$ is the corresponding strength. Classic OT identifies 'deterministic' couplings in that one cell at $t_i$ is transported to a single cell at $t_{i+1}$. Introduction of the entropic regularization term makes this problem nondeterministic, capturing the notion that there may exist immature cells whose fate is not completely determined. Our inferences of multipotentiality is directly a consequence of adding this entropic regularization term. Additionally, entropic regularization also makes the problem strongly convex, which is computationally beneficial.
- The third and the fourth terms are features of unbalanced optimal transport, where equality constraints on the marginals (a consequence of mass conservation) are relaxed. $\lambda_1$ and $\lambda_2$ are the corresponding Lagrange multipliers.

We note that the values of the hyperparameters $\epsilon$, $\lambda_1$, and $\lambda_2$ are held fixed for all pairwise transport map calculations (E13, E14), (E14, E16), ….

## Application of WOT to RGC diversification and long-range couplings

We apply WOT to each pair of consecutive ages $t_i$ and $t_{i+1}$ to estimate the transport map $\hat{\Pi}_{t_i,t_{i+1}}$. Transport maps connecting nonconsecutive time points $t_i$ and $t_{i+k}$ are estimated through a simple matrix multiplication of intermediate transport maps

$$\hat{\Pi}_{t_i,t_{i+k}} = \hat{\Pi}_{t_i,t_{i+1}} \hat{\Pi}_{t_{i+1},t_{i+2}} \dots \hat{\Pi}_{t_{i+k-1},t_{i+k}}$$

The transport matrices $\hat{\Pi}_{t_i,t_j}$ encode fate relationships between cells at $t_i$ and cells at a later time $t_j > t_i$. These relationships can be analyzed at the level of clusters at $t_j$ to associate each cell $\boldsymbol{u}\in\mathcal{A}^{t_i}$ with transcriptomically defined cluster. This is particularly useful in estimating the terminal identity of immature RGCs.

Operationally we compute for each cell $\boldsymbol{u}\in\mathcal{A}^{t_i}$ a 'cell fate vector' $f_{t_j}\left(\beta;\boldsymbol{u}, t_i\right)$, $\left(\beta = 1, 2, \dots\right)$ encoding the probabilities that $\boldsymbol{u}$ is associated with cluster $\mathcal{C}_\beta^{t_j}$ at time $t_j$,

$$f_{t_j}\left(\beta;\boldsymbol{u}, t_i\right) = \frac{\sum_{\boldsymbol{v}\in\mathcal{C}_\beta^{t_j}} \hat{\Pi}_{t_i,t_j}\left(\boldsymbol{u},\boldsymbol{v}\right)}{\sum_\beta \sum_{\boldsymbol{v}\in\mathcal{C}_\beta^{t_j}} \hat{\Pi}_{t_i,t_j}\left(\boldsymbol{u},\boldsymbol{v}\right)}$$

It is easy to verify that

$$\sum_\beta f_{t_j}\left(\beta;\boldsymbol{u}, t_i\right) = 1 \,\forall\, \boldsymbol{u} \in \mathcal{A}^{t_i}$$

The cell fate vector $f_{t_j}\left(\beta; \boldsymbol{u},\, t_i\right)$ encodes probabilistic associations between the cell $\boldsymbol{u}$ and terminal clusters at $t_j > t_i$ indexed by $\beta$. The 'cluster ancestry vector' at an earlier time $t_i$ of a cluster $\mathcal{C}_\beta^{t_j}$ at time $t_j > t_i$, denoted $\Gamma_{t_i}\left(\boldsymbol{u}; \mathcal{C}_\beta^{t_j}\right)$, is defined as follows:

$$\Gamma_{t_i}(\boldsymbol{u}, \mathcal{C}_\beta^{t_j}) = \frac{\sum_{v \in \mathcal{C}_\beta^{t_j}} \hat{\Pi}_{t_i, t_j}(\boldsymbol{u}, \boldsymbol{v})}{\sum_{u \in \mathcal{A}^{t_i}} \sum_{v \in \mathcal{C}_\beta^{t_j}} \hat{\Pi}_{t_i, t_j}(\boldsymbol{u}, \boldsymbol{v})} (t_j > t_i)$$

In a similar vein, the 'cluster descendant vector' at a later time $t_o$ of a cluster $\mathcal{C}_\beta^{t_j}$ at a time $t_j < t_o$, denoted $\Gamma_{t_o}\left(\boldsymbol{u}; \mathcal{C}_\beta^{t_j}\right)$, is defined as

$$\Gamma_{t_o}\left(\boldsymbol{u}; \mathcal{C}_\beta^{t_j}\right) = \frac{\sum_{v \in \mathcal{C}_\beta^{t_j}} \hat{\Pi}_{t_j, t_o}(\boldsymbol{v}, \boldsymbol{u})}{\sum_{u \in \mathcal{A}^{t_o}} \sum_{v \in \mathcal{C}_\beta^{t_j}} \hat{\Pi}_{t_j, t_o}(\boldsymbol{v}, \boldsymbol{u})} \left(t_o > t_j\right)$$

These equations can be used to compute the putative ancestral or descendent cells associated with a cluster $\mathcal{C}_\beta^{t_j}$ at time $t_j$.

## Implementation details of WOT

RGC vectors from all ages were combined, median-normalized, and log-transformed. 2854 HVGs were identified using the Gamma-Poisson model, and WOT was run on this matrix as follows:

```
wot optimal _ transport --matrix RGC _ mat.mtx --cell _ days cell _ day.txt
--growth _ iters 3 --epsilon 0.005 --out tmaps/RGC
```

Cell days were specified in `cell_day.txt` as 0, 1, 3, 6, 11, and 20 for E13, E14, E16, P0, P5, and P56, respectively. We computed trajectories and fates for each age using the following command illustrated for P0:

```
wot trajectory --tmap tmaps/RGC --cell _ set cell _ sets.gmt --day 6 --out
tmaps/traj _ RGC _ P0.txt
```

Fates were computed as

```
wot fates --tmap tmaps/RGC --cell _ set cell _ sets.gmt --day 6 --out tmaps/
fate _ RGC _ P0.txt
```

The above process was repeated for each age.

### Multipotentiality of precursors

For each cell at ages E13–P5, we computed the terminal fate association $f_{P56}\left(\beta; \boldsymbol{u},\, t\right)$, $t \in \left(E13,\, E14,\, \ldots,\, P5\right)$, quantifying the probability that it is a precursor of type $\beta \in \left(1,\, 2,\, \ldots 45\right)$. Note that $f_{P56}\left(\beta; \boldsymbol{u},\, t\right)$ is denoted $f_\beta$ for brevity in the main text. We define

$$P\left(\boldsymbol{u}; t\right) = \frac{1}{\sum_\beta f_{P56}\left(\beta; \boldsymbol{u},\, t\right)^2}$$

as the potential of precursor $\boldsymbol{u}$ at age $t$. Values of $P$ range between 1 and 45, with lower values indicating restriction of fate and higher values suggesting multipotentiality.

## Network analysis of fate couplings

We define

$$C\left(\alpha, \beta; t\right) = \frac{\frac{1}{|\mathcal{A}^t|} \sum\limits_{u \in \mathcal{A}^t} \left(f_{P56}\left(\alpha; \boldsymbol{u},\, t\right) - \overline{f_{P56}\left(\alpha;\, t\right)}\right) \left(f_{P56}\left(\beta; \boldsymbol{u},\, t\right) - \overline{f_{P56}\left(\beta;\, t\right)}\right)}{\sqrt{\frac{1}{|\mathcal{A}^t|} \sum\limits_{u \in \mathcal{A}^t} \left(f_{P56}\left(\alpha; \boldsymbol{u},\, t\right) - \overline{f_{P56}\left(\alpha;\, t\right)}\right)^2} \sqrt{\frac{1}{|\mathcal{A}^t|} \sum\limits_{v \in \mathcal{A}^t} \left(f_{P56}\left(\beta; \boldsymbol{v},\, t\right) - \overline{f_{P56}\left(\alpha;\, t\right)}\right)^2}}$$

as the fate coupling between RGC types $\alpha$ and $\beta$ at age $t$. Clearly, $C\left(\alpha,\beta;t\right)$ is simply the Pearson correlation coefficient between $f_{P56}\left(\alpha;\boldsymbol{u},\,t\right)$ and $f_{P56}\left(\beta;\boldsymbol{u},\,t\right)$, the probabilities that a cell $\boldsymbol{u}\in\mathcal{A}^t$ is a precursor of $\alpha$ and $\beta$ precursor. Here,

$$f_{P56}\left(\alpha;\,t\right) = \frac{1}{|\mathcal{A}^t|}\sum_{\boldsymbol{u}\in\mathcal{A}^t} f_{P56}\left(\alpha;\boldsymbol{u},\,t\right)$$

is the mean probability that a cell at age $t$ is a precursor of type $\alpha$. We computed $C\left(\alpha,\beta;t\right)$ across all 990 pairs of RGC types at each immature age $t\in\left(E13,\,E14,\,\ldots,\,P5\right)$. The values $C\left(\alpha,\beta;E13\right)$ were used as edge weights to visualize the fate coupling network of RGC types using the force-directed layout method (*Fruchterman and Reingold, 1991*) as implemented in the R package igraph. The node layout was computed using $C\left(\alpha,\beta;E13\right)$ values. For other ages, the node layout at E13 was retained but the edges were replotted based on $C\left(\alpha,\beta;t\right)$ values at the corresponding age.

We computed a null distribution of $C\left(\alpha,\beta;t\right)$ by randomizing the values of $f_{P56}\left(\alpha;\boldsymbol{u},\,t\right)$ within each cell $\boldsymbol{u}$ across types. The null values of $C\left(\alpha,\beta;t\right)$ rarely exceeded 0.1 and never exceeded 0.2, so only the edges with larger weights were visualized in *Figure 5*.

## Decay of pairwise couplings

For each pair of RGC types $\alpha$ and $\beta$, we fitted a logistic equation to model the decay of pairwise couplings as

$$C\left(\alpha,\beta;t\right) = \frac{1}{1+\exp\left(\beta_0+\beta_1 t\right)}$$

The values of $t$ corresponding to E13, E14, E16, P0, and P5 were $t$ = 0, 1, 3, 6, and 11, with $C\left(\alpha,\beta;t\right)$ computed as above. We also assumed that $C\left(\alpha,\beta;t\right)=0$ at $t$ = 36, corresponding to P30. Thus, six data points were used to estimate two parameters for each of the 180 pairs of RGC types that had nonzero values of $C\left(\alpha,\beta;t\right)$. The nls function from the R package stats was used to estimate $\beta_0$ and $\beta_1$. The results are plotted in *Figure 5f*.

## Logistic modeling of specification and calculation of $\tau_{sp}$

We hypothesized that the specification of a type $\beta$ corresponds to the localization of its precursors in transcriptomic space. The extent of localization for a RGC of type $\beta$ across the time course was calculated as follows. At each age $t$, we identified the set of precursor RGCs $Prec\left(\beta;t\right)$ showing the highest fate probability corresponding to type $\beta$:

$$Prec\left(\beta;t\right) = \left\{\boldsymbol{u}\in\mathcal{A}^t\,|\,f_{P56}\left(\beta;\boldsymbol{u},t\right) > f_{P56}\left(\alpha\neq\beta;\boldsymbol{u},\,t\right)\right\}$$

Next, we calculated how the precursors of $\beta$ were distributed across clusters at time $t$. We computed the OFs (see above) of precursor cells for type $\beta$ across all clusters $\mathcal{C}_k$, $k$ = 1, 2, $\ldots$, $N\left(t\right)$ at a particular time $t$ ($N\left(t\right)$ is the number of transcriptomically defined clusters at time $t$):

$$p_k\left(\beta;t\right) = \frac{\#\left\{cluster\left(\boldsymbol{u}\right)=\mathcal{C}_k\,\forall\,\boldsymbol{u}\in Prec\left(\beta;t\right)\right\}}{\#\left\{\boldsymbol{u}\in Prec\left(\beta;t\right)\right\}}$$

The localization score for each type $\beta$ at a given time $t$ was defined as

$$Localization\left(\beta;t\right) = 1 - \frac{\sum_{k=1}^{N(t)}\frac{1}{p_k(\beta;t)^2}}{N(t)}$$

where the index $k$ ranges over the number of clusters at time $t$. As defined, $Localization\left(\beta;t\right)$ is restricted to be between 0 and 1, with higher values representing a greater specification. We used a logistic model to approximate the localization of each type as

$$Localization\left(\beta;t\right) = \frac{\exp\left(\gamma_0+\gamma_1 t\right)}{1+\exp\left(\gamma_0+\gamma_1 t\right)}$$

As in the previous section, the nls function was used to estimate the logistic parameters $\gamma_0$ and $\gamma_1$. We consider a type $\beta$ as specified if its specification crosses the line $y\left(t\right) = 0.95\left(1-1/N\left(t\right)\right)$. Thus, the specification time for a type $\beta$ is defined as

$$\tau_{sp}\left(\beta\right) = argmin_t \ Localization\left(\beta; t\right) \geq y\left(t\right)$$

Note that, as defined, $\tau_{sp}$ can be any time point in the interval (E13, P30) corresponding to $t \in (0, 36)$.

## Inference of laterality in RGC types

To identify putative ipsilateral- and contralateral-specified RGC precursors at E13, we scored each precursor RGC based on their expression of bona fide ipsilateral genes (*Zic2*, *Zic1*, and *Igfbp5*) and bona fide contralateral genes (*Isl2*, *Fgf12*, *Igf1*) as in *Wang et al., 2016*. We refer to these as I-RGC and C-RGC scores. Putative I-RGCs were those cells that expressed the I-RGC score at 1.5 standard deviations higher than the mean across all cells, and those that express the C-RGC score at 1.5 standard deviations lower than the mean across all cells. C-RGCs were defined analogously. Many cells did not express either of these marker sets as shown in *Figure 7c*. These are likely to be RGCs that have not declared their laterality or C-RGCs that are not defined by the expression of *Isl2*, *Fgf12*, and *Igf1*.

WOT was then used to compute the descendants of E13 I-RGCs at all subsequent ages through P56 using the `wot fates` command introduced above. These descendants were used for two purposes. First, we assessed the proportion of putative I-RGCs across types as in *Figure 7d*. We also performed a differential gene expression test between putative I-RGCs and the remaining RGCs at all ages, as shown in *Figure 7e and f* and *Figure 7—figure supplement 1d and e*.

## Comparison with previous retina scRNA-seq datasets

We compared our data with three scRNA-seq studies that profiled the whole retina during development:

1. (*Clark et al., 2019*): Count matrices and cell-/gene-level annotations were downloaded from the author's public repository https://github.com/gofflab/developing_mouse_retina_scRNASeq (*Goff Lab, 2021*). This dataset contains whole retinal cells sampled at 10 time points (E11, E12, E14, E16, E18, P0, P2, P5, P8, P14) with four of these (E14, E16, P0, P5) common with our study. We excluded P5 from our analysis as only N = 11 RGCs were identified by the authors at this time point.
2. (*Lo Giudice et al., 2019*): Count matrix corresponding to E15.5 retinal cells was kindly provided by the authors.
3. (*Rheaume et al., 2018*): Count matrix corresponding to P5 RGCs was downloaded from the online submission.

For consistency with our filtering parameters, we extracted cells based on a cutoff of 700 genes/cell from each of the above datasets. For the Clark et al. dataset, this selected 17,827 cells at E14, 1674 cells at E16, and 8343 cells at P0, respectively (N = 27,844 cells). In these samples, RGCs comprised 19, 28, and 0.45%. For the Giudice et al. dataset, this selected 5218 cells, of which 23% were RGCs.

The Rheaume et al. dataset was directly compared with P5 RGCs collected in this study using supervised classification (*Figure 1—figure supplement 1i*). Clark et al. and Lo Giudice et al. data were combined with the retinal cells profiled in this study at corresponding time points (25,685 cells at E14; 21,274 cells at E16; and 23,251 cells at P0). Together, this resulted in a 14,350 genes × 103,272 cells expression matrix that was analyzed following the steps outlined previously. In the alignment step, cells from each combination of age and study were considered as a separate 'batch.' We visualized the transcriptional heterogeneity of the full dataset using UMAP and used the expression of canonical markers to confirm the co-clustering of cell classes in *Figure 1—figure supplement 1* (*Rbpms* for RGCs, *Tfap2b* for ACs, *Fgf15* for RPCs, and *Gngt2* for RPCs).

## Data availability

All scRNA-seq data collected in this study were submitted to the Gene Expression Omnibus (GEO) under GSE185671. The data can be visualized on the Broad Institute's Single Cell Portal under the identifier SCP1706.

## Code availability

The scripts (written in R) generated for this study are shared at https://github.com/shekharlab/mouseRGCdev, (copy archived at swh:1:rev:ca6a97adabb7bc4ffb2fb1187c78cb277513665c; *Shekhar, 2022*).

## Acknowledgements

Funding for this study was provided by the National Institute of Health grants R37NS029169, R01EY022073 (JRS), NSF GRP DGE1752814 (SB), and R00EY028625 (KS), and startup funding from UC Berkeley (KS, SB). We are grateful to Prof. Seth Blackshaw for helpful suggestions on the preprint. We thank Joshua Hahn for valuable feedback.

## Additional information

### Funding

| Funder | Grant reference number | Author |
|---|---|---|
| National Institutes of Health | R37NS029169 | Joshua R Sanes |
| National Institutes of Health | R01EY022073 | Joshua R Sanes |
| National Institutes of Health | R00EY028625 | Karthik Shekhar |
| National Science Foundation | GRP DGE1752814 | Salwan Butrus |

The funders had no role in study design, data collection and interpretation, or the decision to submit the work for publication.

### Author contributions

Karthik Shekhar, Conceptualization, Data curation, Formal analysis, Funding acquisition, Investigation, Methodology, Project administration, Resources, Software, Supervision, Validation, Visualization, Writing – original draft, Writing – review and editing; Irene E Whitney, Conceptualization, Formal analysis, Investigation, Methodology, Resources, Validation, Writing – review and editing; Salwan Butrus, Formal analysis, Software, Visualization, Writing – review and editing; Yi-Rong Peng, Investigation, Methodology, Writing – review and editing; Joshua R Sanes, Conceptualization, Formal analysis, Funding acquisition, Methodology, Project administration, Supervision, Visualization, Writing – original draft, Writing – review and editing

### Author ORCIDs

Karthik Shekhar (iD) http://orcid.org/0000-0003-4349-6600
Joshua R Sanes (iD) http://orcid.org/0000-0001-8926-8836

### Ethics

All animal experiments were approved by the Institutional Animal Care and Use Committees (IACUC) at Harvard University. Mice were maintained in pathogen-free facilities under standard housing conditions with continuous access to food and water. Animals used in this study include both males and females. A meta-analysis (not shown) did not show any systematic sex-related effects in either differentially expressed genes or cell-type proportions.

### Decision letter and Author response

Decision letter https://doi.org/10.7554/eLife.73809.sa1
Author response https://doi.org/10.7554/eLife.73809.sa2

## Additional files

### Supplementary files

• Transparent reporting form

## Data availability

Sequencing data has been submitted under GSE185671. Reviewer token : evchicgutpqpnoj. Computational scripts are available at: https://github.com/shekharlab/mouseRGCdev, (copy archived at swh:1:rev:ca6a97adabb7bc4ffb2fb1187c78cb277513665c).

The following dataset was generated:

The following previously published dataset was used:

| Author(s) | Year | Dataset title | Dataset URL | Database and Identifier |
|---|---|---|---|---|
| Tran NM, Shekhar K | 2019 | Single-Cell Profiles of Retinal Ganglion Cells Differing in Resilience to Injury Reveal Neuroprotective Genes | https://www.ncbi.nlm.nih.gov/geo/query/acc.cgi?acc=GSE137400 | NCBI Gene Expression Omnibus, GSE137400 |

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
