## [Editor Report]

Your study using single-cell RNA-seq to profile developing retinal ganglion cells from embryonic and postnatal mouse retina showcases the diversification of this class of neuron into specific subtypes. The computational approaches you developed identify groups of RGC precursors with largely nonoverlapping fates, distinguished by selectively expressed transcription factors that could act as fate determinants. You then show that over time clusters of cells become ‘decoupled’ as they split into subclusters, indicating that subtype diversification arises as a gradual, asynchronous fate restriction of postmitotic multipotential precursors. Your data should enable the neural development community to generate new hypotheses in the field of retinal ganglion cell differentiation and beyond in other neural structures.

---

## [Decision Letter]

**Decision letter after peer review:**

Thank you for submitting your article "Diversification of multipotential postmitotic mouse retinal ganglion cell precursors into discrete types" for consideration by *eLife*. Your article has been reviewed by 2 peer reviewers, and the evaluation has been overseen by a Reviewing Editor and Claude Desplan as the Senior Editor. The following individual involved in review of your submission has agreed to reveal their identity: Tom Reh (Reviewer #1).

The reviewers had several questions and comments that could be addressed textually or by delving further into existing data; these fall into five aspects:

1) The variability among the cells at the early time points may be due to their different stages of differentiation, since they are a composite of cells with different birthdates. This point should be noted and discussed, or you can try to tease out this aspect from the data.

2) The "differential" between ipsilateral and contralateral RGCs – timing of birth, and genes expressed.

3) The relationships among the cells are not true lineages, but inferred lineages. Reviewer 2 called for an additional method to confirm these potential relationships for at least a some of the subtypes, by retrograde tracing in postnatal stages to cull out the ipsilateral cells. Retrograde labeling is difficult past late embryonic stages. While genetic lineage tracing is beyond the scope of this paper, you could consider making use of the Sert-Cre mouse, which labels the ipsilateral RGCs in the ventrotemporal retina, in conjunction with validation of specific genes by in situ hybridization.

4) Describing the methods in a clearer, more accessible way.

5) Comparing your findings with the several recent studies on transcriptomics of the retina (e.g., Lo Giudice, Rheaume, etc..).

*Reviewer #1 (Recommendations for the authors):*

While the data presentation is very clear and data support the overall conclusions, there are a few points the authors could consider.

1. The authors initially use standard clustering approaches to determine when in development RGCs form distinct clusters. As early as E13, 10 distinct clusters were present (Figure 2B). It is not shown, however, how these clusters relate to those they define with the WOT analysis that follows. A similar UMAP plot in Figure 4d, for example, suggests that clusters 8 and 9 have less potential than cluster 6. It would be useful to see how the clustering and WOT analyses relate.

2. The authors turn to WOT analysis to examine the relationships among cells within clusters. They state " Thus, WOT identifies fate associations between individual cells". This seems like an overstatement, since to my understanding methods like WOT can only infer relationships among cells based on their degree of transcriptomic similarity.

3.The authors find that over time, clusters of cells become "decoupled" as they split into subclusters. This process of fate decoupling is associated with changes in the expression of specific transcription factors. This allows them to both predict lineage relationships among RGC subtypes and the time during development when these specification events occur.

It would be nice if there was a single RGC type defined by an orthogonal method that showed these features of progressive restriction that corresponded with the computational prediction. For example, the senior author reported several years ago that the Cdh6+ RGCs are specified very early in the differentiation process (Huetra et al., 2012). This would suggest that not all subtypes are multipotential.

4. The authors also show that "early specification of the Eomes group is consistent with birthdating studies showing the average earlier birthdate of ipRGCs compared to all RGCs (McNeill et al., 2011)." It would be useful to show whether this is a more general feature of the data analysis. For example, Osterhout et al., 2014, has shown that Cdh3, Drd4 and Hoxd10 RGC subtypes have quite distinct periods of genesis. If these cell types can be identified as members of clusters at the relevant ages, the authors might be able to determine the relationship between birthdate and decoupling/specification.

5. Another feature of the RGCs at each age is that some of these were "born yesterday" while other may have been around for several days since they were generated. Therefore, there should be a gradient of RGC maturation, particularly at the later ages. It is not clear how this would affect the conclusions. I can imagine for example, that the degree of maturation of the RGC might affect the estimate of specification of the cell type in the overall population.

*Reviewer #2 (Recommendations for the authors):*

Please find additional questions and comments, as well as suggestions for the authors:

– Is it possible to validate with more direct methods (instead of inference) the observation of a late diversification? Some efficient method to address this question have been described before, such as lineage tracing technique, either viral or CRISPR based methodology (eg. by Schier lab: DOI: 10.1038/nbt.4103; Junker lab; doi: 10.1038/nbt.4124 and Morris lab: 10.1038/s41586-018-0744-4 ; also reviewed by McKenna and Gagnon: https://doi.org/10.1242/dev.169730).

– The study is very descriptive and may benefit from an experimental set up showing this framework is related to functionally relevant features. As the stages chosen by the authors are based on (i) peak of RGC generation (E13/E14); (ii) peak of RGC axons reaching target and (iii) dendrite arborization. Would it be possible to challenge one of these processes to assess whether axon guidance choices or synaptic activity are important to instruct the diversity bursts that is observed after P0? (i.e., support with a functional perturbation the mere observations).

– The authors discuss the presence of transient (DV) ipsilateral RGC (Soares and Mason, 2015) and decide to analyze the non- transient population (Zic2+). Is this population present at E13?

– The I-RGC analysis is mostly based on the inference of their identity based on a few markers. To make a strong point, I-RGC shall be isolated at a stage of final maturation (P5?) and using a robust method such as retrograde labelling.

– RGC are known to undergo a significant wave of cell death between E16 and P5. How much the loss of this pool RGC could be associated in the increase in discreetness observed over time? Did the analysis take that into account?

---

## [Author Response]

The reviewers have discussed their reviews with one another, and the Reviewing Editor has drafted this to help you prepare a revised submission.The reviewers had several questions and comments that could be addressed textually or by delving further into existing data; these fall into five aspects:1) The variability among the cells at the early time points may be due to their different stages of differentiation, since they are a composite of cells with different birthdates. This point should be noted and discussed, or you can try to tease out this aspect from the data.

We agree that an important axis of transcriptomic variation in our data, especially at early stages, may be due to stages of differentiation associated with different birthdates (asynchrony). We considered the possibility of inferring birthdate based on transcriptional profiles, but were unsuccessful. For example, we searched for gradients of expression of RGC class markers within the immature clusters as a potential way to address this issue. However, we found no substantially greater inter-cluster variation at early times than in adults. We now mention this in the discussion.

2) The "differential" between ipsilateral and contralateral RGCs – timing of birth, and genes expressed.

We include additional discussion of the ways in which we defined ipsilateral and contralateral RGCs. We also re-emphasize that these are tentative identification, based on molecular markers obtained by other groups, primarily that of Carol Mason.

3) The relationships among the cells are not true lineages, but inferred lineages. Reviewer 2 called for an additional method to confirm these potential relationships for at least a some of the subtypes, by retrograde tracing in postnatal stages to cull out the ipsilateral cells. Retrograde labeling is difficult past late embryonic stages. While genetic lineage tracing is beyond the scope of this paper, you could consider making use of the Sert-Cre mouse, which labels the ipsilateral RGCs in the ventrotemporal retina, in conjunction with validation of specific genes by in situ hybridization.

We agree with this point and apologize if our attempts to acknowledge it fell short. Unfortunately, the suggested experiments are outside the scope of this paper as they would at least take a year for setup and analyses. This reflects not only their difficulty but also the sad fact that we are not currently equipped to perform them in our laboratories. Instead, we have added additional text to emphasize that these are inferred rather than true lineages.

4) Describing the methods in a clearer, more accessible way.

We have rearranged and edited the methods section in a number of places as requested to improve readability. We note, however, that a full description of the computational methods requires inclusion of mathematical detail that is not going to be accessible to those without some specialized knowledge.

5) Comparing your findings with the several recent studies on transcriptomics of the retina (e.g., Lo Giudice, Rheaume, etc..).

We had included comparisons with Lo Giudice et al. and Clark et al. in Figure S1e-h. As described, the main difference between our study and theirs is that we isolated RGCs, which are a minority subclass. Consequently, Lo Giudice et al. and Clark et al. lacked the resolution to resolve RGC diversity at later time points. Regarding Rheaume et al., who isolated and profiled RGCs at P5, we had cited that paper in several places and have now added a comparison between their and our P5 datasets, showing that both identified nearly identical types. Finally, we note that the concordance of RGC types isolated with two different markers supports the idea that we have not missed types.

Reviewer #1 (Recommendations for the authors):While the data presentation is very clear and data support the overall conclusions, there are a few points the authors could consider.1. The authors initially use standard clustering approaches to determine when in development RGCs form distinct clusters. As early as E13, 10 distinct clusters were present (Figure 2B). It is not shown, however, how these clusters relate to those they define with the WOT analysis that follows. A similar UMAP plot in Figure 4d, for example, suggests that clusters 8 and 9 have less potential than cluster 6. It would be useful to see how the clustering and WOT analyses relate.

We had performed an analysis of the type requested; results are presented in Figure 4—figure supplement 2, where the fate relationships inferred by WOT among individual cells is collapsed to a “cluster-cluster” association map similar to that produced by supervised classification in Figure 3. We find that the results are highly concordant – at all time points the Pearson correlation coefficients of the cluster-cluster associations between the two methods is greater than 0.9. With regards to the specific example of cluster 8 and 9 at E13, their pattern of association with E14 clusters is similar between Figure 3d (supervised classification) and Figure 4—figure supplement 2a (WOT). Thus, the two methods are consistent at the cluster level. We have noted this in the main text (lines 368-370).

In addition to the fact that it acts on cells rather than clusters, WOT provides long-range fate associations that allow us to directly relate each precursor to a terminal type (illustrated in Figure 4). It is for this reason that we used WOT for some subsequent analyses in the paper after demonstrating its concordance with the classification results at the level of clusters.

2. The authors turn to WOT analysis to examine the relationships among cells within clusters. They state " Thus, WOT identifies fate associations between individual cells". This seems like an overstatement, since to my understanding methods like WOT can only infer relationships among cells based on their degree of transcriptomic similarity.

We apologize for the confusion. WOT, like all methods that infer developmental processes from single-cell RNA-seq data, relies on transcriptomic similarity as a proxy for fate proximity. Between the two methods used in this study, supervised classification and WOT, the key difference is that supervised classification relies on comparing similarity only at the level of pre-determined clusters. Thus, supervised classification yields a measure of transcriptomic similarity (i.e. fate association) between a cluster at a given time point and every cluster at a consecutive time point. In contrast, WOT computes fate associations between every *cell* at a given time point and *every cell* at a consecutive time point. The outcome is that for each cell at E14 (for example) we have a vector of fate associations with respect to each cell at E16 as well as each cell at E13.

While WOT results can be interpreted at the level of clusters *post hoc*, there is no principled way to resolve cellular relationships from the results of supervised classification. We have modified the wording in Results (line 365-366) to avoid giving the wrong impression, and added a form of the explanation above to the Discussion (line 748-749).

3.The authors find that over time, clusters of cells become "decoupled" as they split into subclusters. This process of fate decoupling is associated with changes in the expression of specific transcription factors. This allows them to both predict lineage relationships among RGC subtypes and the time during development when these specification events occur.It would be nice if there was a single RGC type defined by an orthogonal method that showed these features of progressive restriction that corresponded with the computational prediction. For example, the senior author reported several years ago that the Cdh6+ RGCs are specified very early in the differentiation process (Huetra et al., 2012). This would suggest that not all subtypes are multipotential.

We agree that validation with a second method would be useful. Unfortunately, additional experiments using genetically engineered mice are outside the scope of this paper as they would at least take a year for setup and analyses. This reflects not only their difficulty but also the sad fact that we are not currently equipped to perform them in our laboratories. For example, we no longer maintain either the Cdh6-cre line or the Sert-cre line mentioned elsewhere. Instead, we have added additional text in a number of places to emphasize that these are inferred rather than true lineages.

Regarding the de la Huerta paper, we admit to ignorance and befuddlement. Cdh6 is expressed in multiple clusters at early stages but at exceedingly low levels (<2% of cells; Author response image 1) to be considered robust. We are therefore unable to identify *Cdh6+* precursor RGCs and examine their fate *in silico*. Since very low levels of *cre* expression can activate reporters it may be that the relevant levels of endogenous *Cdh6* are too low to be detected by the somewhat shallow sequencing of the scRNA-seq method.

**Author response image 1. sa2fig1:** Low levels of expression of Cdh6 in immature clusters at E13 (left) and E16 (right). Cdh6 is expressed at similarly low levels at all other time points (not shown). RGC markers Pou4f2 and Rbpms are shown for comparison.

4. The authors also show that "early specification of the Eomes group is consistent with birthdating studies showing the average earlier birthdate of ipRGCs compared to all RGCs (McNeill et al., 2011)." It would be useful to show whether this is a more general feature of the data analysis. For example, Osterhout et al., 2014, has shown that Cdh3, Drd4 and Hoxd10 RGC subtypes have quite distinct periods of genesis. If these cell types can be identified as members of clusters at the relevant ages, the authors might be able to determine the relationship between birthdate and decoupling/specification.

This is a good idea but unfortunately these markers and lines cannot be used. For all three lines, there is a discrepancy between the cells that are labeled and expression of the endogenous gene. We obtained the *Cdh3-GFP* line to which the reviewer refers many years ago and studied it in some detail, but did not publish on it. It turns out that the line labels several different types of RGCs, of which many (perhaps most) of which don’t actually express *Cdh3*. Similarly, we showed several years ago that labeled cells in the *Drd4-GFP* line do not express detectable *Drd4* (Kay…Sanes, J. Neuroscience, 2011). We rechecked both genes in our new datasets and found very low levels of expression at all time points. We have no experience with the *Hoxd10-GFP* line but found no Hoxd10 expression at all in our dataset. We and others have previously discussed reasons why BAC transgenic lines such as these frequently lead to such “ectopic” expression.

5. Another feature of the RGCs at each age is that some of these were "born yesterday" while other may have been around for several days since they were generated. Therefore, there should be a gradient of RGC maturation, particularly at the later ages. It is not clear how this would affect the conclusions. I can imagine for example, that the degree of maturation of the RGC might affect the estimate of specification of the cell type in the overall population.

Thank you for raising this important point. At any of the stages profiled, the degree of maturation is likely to be an important axis of variation that is intermixed with transcriptional signatures of type-identity. However, in contrast to your expectation that it should be important at later stages, we believe this is likely important at earlier stages in the data. Empirically, the clusters defined at P5 exhibit highly specific correspondences to P56 clusters, where we assume that maturational gradients are absent. This suggests that the predominant axis of transcriptional variation at P5 is likely type-identity. Thus, although cells within a type-defining cluster may exhibit maturational differences, these are unlikely to impact any of the analysis or conclusions.

In contrast, these effects could be important at early stages where clusters are not very well separated, and the most multipotential precursors are intermixed with cells that are at advanced stages of commitment. Here, it is very much possible that some clusters are defined by maturational state rather than type-identity. However, we were unable to detect such patterns using the relevant transcriptional signatures. As both methods (supervised classification and WOT) connect clusters/cells across time points based on transcriptional proximity, less mature precursors and more mature precursors are likely to map to their counterparts across time points. Moreover, neither method forcefully maps every cellular state across time points. Thus, if multipotential RGC precursors are not present at time (t+1), the corresponding cells at time t do not contribute any forward weight.

Reviewer #2 (Recommendations for the authors):Please find additional questions and comments, as well as suggestions for the authors:– Is it possible to validate with more direct methods (instead of inference) the observation of a late diversification? Some efficient method to address this question have been described before, such as lineage tracing technique, either viral or CRISPR based methodology (eg. by Schier lab: DOI: 10.1038/nbt.4103; Junker lab; doi: 10.1038/nbt.4124 and Morris lab: 10.1038/s41586-018-0744-4 ; also reviewed by McKenna and Gagnon: https://doi.org/10.1242/dev.169730).

We believe that additional experiments of this type, while important and interesting, are beyond the scope of this study. Our reasoning is detailed above, and we note that we have checked with the editors on this point and received their assurance that additional experiments will not be required for acceptance.

– The study is very descriptive and may benefit from an experimental set up showing this framework is related to functionally relevant features. As the stages chosen by the authors are based on (i) peak of RGC generation (E13/E14); (ii) peak of RGC axons reaching target and (iii) dendrite arborization. Would it be possible to challenge one of these processes to assess whether axon guidance choices or synaptic activity are important to instruct the diversity bursts that is observed after P0? (i.e., support with a functional perturbation the mere observations).

We believe that functional perturbations experiments of this type are beyond the scope of this study. We are currently not set up to do experiments of the kind suggested. Our reasoning is detailed above, and we note that we have checked with the editors on this point and received their assurance that additional experiments will not be required for acceptance.

– The authors discuss the presence of transient (DV) ipsilateral RGC (Soares and Mason, 2015) and decide to analyze the non- transient population (Zic2+). Is this population present at E13?

We spent some time trying to find a way to identify transient I-RGCs and unfortunately found none. To our knowledge reliable molecular markers are not available to identify transient I-RGCs at E13.

– The I-RGC analysis is mostly based on the inference of their identity based on a few markers. To make a strong point, I-RGC shall be isolated at a stage of final maturation (P5?) and using a robust method such as retrograde labelling.

We believe that additional experiments of this type are beyond the scope of this study. Our reasoning is detailed above, and we note that we have checked with the editors on this point and received their assurance that additional experiments will not be required for acceptance.

– RGC are known to undergo a significant wave of cell death between E16 and P5. How much the loss of this pool RGC could be associated in the increase in discreetness observed over time? Did the analysis take that into account?

We spent some time trying to find a way to analyze whether naturally occurring cell death acts disproportionately on some RGC types or on RGCs that remain multipotential. Unfortunately, we found no way to address this issue.